# EMPIRICALLY INVESTIGATING THE TRADE-OFFS IN DETERMINISTIC CERTIFIED TRAINING

## ABSTRACT

While there have been numerous advancements regarding the performance of deep neural networks on a broad range of supervised learning tasks , their adversarial robustness remains a major concern. To mitigate this, *neural network verification* aims to provide mathematically rigorous robustness guarantees at the cost of substantial computational requirements. *Certified training* methods overcome this challenge by optimising for verifiable robustness during training, which, however, usually results in substantial decrease of performance on clean data. This *robustness-accuracy* trade-off has been extensively studied in the context of adversarial training but remains mostly unexplored for certified training. To control this trade-off, certified training techniques expose hyperparameters, which, to date, have been manually tuned to one specific configuration that compares favourable to the previous state-of-the-art. In this work, we present a novel fully-automated hyperparameter optimisation procedure for certified training that yields a Pareto front of optimal configurations with regard to the robustness-accuracy trade-off. Our approach facilitates the fair, principled and nuanced comparison of the performance of different methods. We show that most methods yield better trade-offs than previously assumed, thereby establishing a new state of the art in certified training of deep neural networks. In addition, we demonstrate that performance improvements reported over recent years are far less pronounced when all methods have been carefully tuned.

## 1 INTRODUCTION

In recent years, deep learning has enabled remarkable advances across several application areas ranging from computer vision (Dosovitskiy et al., 2021) to protein structure prediction (Jumper et al., 2021). Concurrently, there has been a fast-growing trend towards employing deep-learning-based systems in safety-critical domains, such as unmanned aircraft manoeuvre advisory systems (Julian et al., 2019) and map generation for autonomous driving (Hubbertz et al., 2025). However, it is well known that deep neural networks are vulnerable to *adversarial examples* (Szegedy et al., 2014): inputs perturbed by small, carefully designed modifications that lead to misclassification (see, *e.g.*, Goodfellow et al. (2015); Madry et al. (2018)).

While adversarial attacks play an important role in diagnosing weaknesses before or after deployment, because of the heuristic nature of the methods and their reliance on local gradients, they may fail to find an adversarial manipulation of given inputs even when those exist. Thus, *neural network verification* techniques have been proposed that provide *formal guarantees* on the robustness of neural networks (see, *e.g.*, Tjeng et al. (2019); Wang et al. (2021); Ferrari et al. (2022); De Palma et al. (2024a)). These come at the cost of substantially increased computational requirements, since proving even simple properties is an $\mathcal{NP}$-complete task (Katz et al., 2017; Sälzer & Lange, 2021).

One commonly studied property in the context of neural network verification is local robustness within an $\ell_\infty$ norm-ball around inputs (see, *e.g.*, Wang et al. (2021); Brix et al. (2023); König et al. (2024)). To train networks that adhere to that property, several techniques have been proposed, most prominently *adversarial training* (see, *e.g.*, Madry et al. (2018); Zhang et al. (2019)). Here, the parameters of the neural network are optimised with regard to a worst-case loss within the given threat model approximated by means of adversarial attacks.

While these techniques result in neural networks that are empirically robust, the resulting networks are usually not easily-verifiable, *i.e.*, even highly-optimised state-of-the-art solvers mostly fail to prove robustness properties (see, *e.g.*, Mao et al. (2025); De Palma et al. (2024b)). An orthogonal line of research, *certified training*, focuses on producing networks for which formal robustness guarantees can be obtained more efficiently (see, *e.g.*, Gowal et al. (2019); Zhang et al. (2020); Shi et al. (2021); Müller et al. (2023); Mao et al. (2023); De Palma et al. (2024b)). Here, *incomplete verification* methods that yield sound, but potentially loose, bounds on the outputs of the neural network are employed to over-approximate the worst-case loss.

State-of-the-art methods rely on *Interval Bound Propagation* (IBP) (Gowal et al., 2019) for the bounding process. While the resulting networks are more amenable to formal verification techniques, compared to adversarially trained networks, they generally perform far worse on clean data (see, *e.g.*, Müller et al. (2023); De Palma et al. (2024b)). This effect is known as the *robustness-accuracy trade-off* in the context of adversarial training (see, *e.g.*, Tsipras et al. (2019); Zhang et al. (2019)), but remains mostly unexplored for deterministic certified training methods.

State-of-the-art certified training techniques expose hyperparameters that govern the trade-off between robustness and accuracy. In particular, they introduce a weighting factor to balance the certified loss obtained through IBP against either clean loss (Gowal et al., 2019; Zhang et al., 2020) or adversarial loss (Müller et al., 2023; De Palma et al., 2024b). Moreover, these methods require tuning additional hyperparameters, such as the learning rate and the number of warm-up epochs, which strongly influence training stability and final performance. Until now, the state of the art in certified training has been determined by tuning methods to one specific trade-off that improves over results from related work; manually (see, *e.g.*, Müller et al. (2023); De Palma et al. (2024b)) or by relying on grid search (Mao et al., 2025). However, due to the robustness-accuracy trade-off, the problem naturally gives rise to a *Pareto front* of configurations, i.e., a set of configurations for which improving one objective necessarily degrades the other. To date, this front has not been systematically explored in the context of certified training.

In this work, we propose, for the first time, a method for computing a Pareto front of well-performing hyperparameter configurations of certified training techniques with regard to natural and certified accuracy by employing methods from the field of multi-objective hyperparameter optimisation. However, these methods cannot be trivially applied to certified training. Assessing the final target objective, *i.e.*, the certified robustness of a network obtained via complete verification, for each investigated configuration is infeasible. We demonstrate that an estimation of certified robustness computed through cheaper, incomplete verification techniques serves as an efficient proxy objective, yielding networks that also perform well under complete verification. Furthermore, certain regions of the Pareto front correspond to trivial configurations; for example, the highest natural accuracies can be obtained by training solely on clean or adversarial loss respectively. To avoid expending resources on these regions, we demonstrate how the optimisation can be effectively constrained to focus only on interesting areas.

To summarise, our contributions are as follows:

1. We introduce the first fully automated hyperparameter optimisation framework for certified training based on constrained multi-objective optimisation, which computes a Pareto front of optimal configurations, balancing performance and verifiability.

2. Using this framework, we demonstrate that many existing certified training methods achieve more favourable trade-offs than previously reported across standard benchmarks, thereby establishing a new state of the art in certified training.

3. Lastly, we show how a more nuanced assessment of the state of the art in certified training is enabled by the computed Pareto fronts, revealing complimentary performance between methods when higher certified or clean accuracies are desired.

## 2 BACKGROUND

In the following, we provide the necessary background for our work, covering neural network verification, certified training and multi-objective hyperparameter optimisation.

## 2.1 Neural Network Verification

Generally, given a neural network $f_\theta : \mathbb{R}^d \mapsto \mathbb{R}^c$, $c, d \in \mathbb{N}$ that maps inputs $\mathbf{x} \in \mathbb{R}^d$ to outputs $f_\theta(\mathbf{x}) \in \mathbb{R}^c$, *formal neural network verification* is concerned with proving whether a given input-output property *holds* or *is violated* for $f$.

In this study, we focus on classification problems with scalar labels $y \in \mathbb{N}$ and on *local robustness* in an $\ell_\infty$ norm ball with radius $\epsilon$ denoted as $\mathcal{B}_\infty^\epsilon$. More formally, given an original input $x_0$ with correct label $y_0$, the local robustness problem can be stated as

$$\forall \mathbf{x}' \in \mathcal{B}_\infty^\epsilon := \{\mathbf{x} \mid ||\mathbf{x} - \mathbf{x}_0||_\infty \leq \epsilon\} : \arg\max_j f_\theta(\mathbf{x}')_j = y_0 \tag{1}$$

The problem reduces to computing the sign of the following optimisation problem, where $\mathbf{z}(\mathbf{x}, y) \in \mathcal{R}^c$ is defined as the vector of logit differences, *i.e.*, $\mathbf{z}(\mathbf{x}, y) := f_\theta(\mathbf{x})[y] \cdot \mathbf{1} - f_\theta(\mathbf{x})$:

$$\min_{\mathbf{x}' \in \mathcal{B}_\infty^\epsilon} \min_{i \neq y} \mathbf{z}(\mathbf{x}', y)[i] \tag{2}$$

Computing an exact solution to Equation 2 is known to be an $\mathcal{NP}$-complete problem (Sälzer & Lange, 2021; Katz et al., 2017). Therefore, in practice, sound lower bounds $\underline{\mathbf{z}}(\mathbf{x}, y)[i] \leq \mathbf{z}(\mathbf{x}, y)[i], i \in \{1, \ldots, c\}$ are approximated using *incomplete* verification methods.

The arguably conceptually simplest incomplete method is *Interval Bound Propagation* (IBP) (Gowal et al., 2019; Mirman et al., 2018), which employs axis-aligned hyper-boxes to approximate the set of possible outputs. For this, consider $f_\theta$ as the composition of $L$ linear layers $h_{1,\ldots,L}$ with $h_i(\mathbf{x}^{i-1}) = \mathbf{W}_i \cdot \mathbf{x}^{i-1} + \mathbf{b}_i$ and the ReLU activation $\sigma(\mathbf{x}) := \max(0, \mathbf{x})$, *i.e.*, $f_\theta = h_L \circ \sigma \circ h_{L-1} \circ \cdots \circ \sigma \circ h_1$. Using interval arithmetic, the axis-aligned hyper-box $\mathcal{B}_1$ that encompasses $h_1(\mathcal{B}_\infty^\epsilon)$ is defined to have centre $\overline{\mathbf{x}}_1 = \mathbf{W} \cdot \mathbf{x}_0$ and edge length $\delta_1 = |\mathbf{W}| \cdot \epsilon$. To approximate the reachable outputs of $\sigma(\mathcal{B}_1)$, due to the non-linearity of the ReLU function, lower and upper bounds have to be propagated separately, *i.e.*, $\mathbf{l}_2 = \sigma(\overline{\mathbf{x}}_1 - \delta_1)$ and $\mathbf{u}_2 = \sigma(\overline{\mathbf{x}}_1 + \delta_1)$. The resulting hyper-box $\mathcal{B}_2$ has centre $\overline{\mathbf{x}}_2 = \frac{\mathbf{u}_2 + \mathbf{l}_2}{2}$ and edge length $\delta_2 = \frac{\mathbf{u}_2 - \mathbf{l}_2}{2}$. By continuing this process, we can compute a hyper-box that encompasses the reachable output set of $f_\theta$, thereby allowing for the calculation of $\underline{\mathbf{z}}(\mathbf{x}, y)$.

More sophisticated methods, such as $(\alpha\text{-})$CROWN (Zhang et al., 2018; Xu et al., 2021), propagate symbolic intervals and employ a tighter relaxation at the cost of increased computational complexity. Furthermore, incomplete methods can be used within a branch-and-bound framework (Bunel et al., 2020) that solves the verification problem in a complete fashion (see, *e.g.*, De Palma et al. (2024a); Ferrari et al. (2022); Wang et al. (2021)). These methods constitute the current state of the art in complete neural network verification (Brix et al., 2024; König et al., 2024).

## 2.2 Training Robust Neural Networks

Madry et al. (2018) introduced the problem of training robust neural networks as a min-max optimisation problem that aims to find parameters $\theta$ that minimise an expected worst-case loss measured through $\mathcal{L} : \mathbb{R}^c \times \mathbb{N} \to \mathbb{R}$ in the $\ell_\infty$ norm ball around samples from a data distribution $(\mathbf{x}, y) \sim D$:

$$\theta \in \arg\min_{\theta'} \mathbb{E}_D[\max_{\mathbf{x}' \in \mathcal{B}_\infty^\epsilon} \mathcal{L}(f_{\theta'}(\mathbf{x}'), y)] \tag{3}$$

As mentioned previously, calculating the exact worst-case loss is computationally not feasible, since it is equivalent to solving Equation 2. Therefore, Madry et al. (2018) under-approximate the inner maximisation by means of *Projected Gradient Descent* (PGD), which iteratively searches for points $\mathbf{x}_{\text{adv}}$ in $\mathcal{B}_\infty^\epsilon$ that maximise the worst-case loss. We refer to this as the *adversarial loss* $\mathcal{L}_{\text{adv}} := \mathcal{L}(f_\theta(\mathbf{x}_{\text{adv}}), y)$. While the resulting networks are empirically robust, *i.e.*, far more resistant to adversarial attacks than traditionally trained networks, they do not yield certifiable guarantees and may be vulnerable to stronger adversarial attacks (Mao et al., 2025; Croce et al., 2021). *Certified training* methods follow an orthogonal approach by over-approximating the true value of the inner maximisation by means of incomplete verification methods. The *verified loss* $\mathcal{L}_{\text{ver}}$ is computed on the previously defined lower bound to the logit differences of $f_\theta$ (Wong & Kolter, 2018):

$$\max_{\mathbf{x}' \in \mathcal{B}_\infty^\epsilon} \mathcal{L}(f_\theta(\mathbf{x}'), y)] \leq \mathcal{L}_{\text{ver}} := \mathcal{L}(-\underline{\mathbf{z}}(\mathbf{x}, y), y) \tag{4}$$

This loss decreases when the employed incomplete verifier can prove that $f_\theta$ is locally robust for the given training sample. Perhaps suprisingly, training methods that employ the hyper-box relaxation currently yield best results, despite relying on a relatively loose over-approximation (see, *e.g.*,

De Palma et al. (2024b); Mao et al. (2024); Müller et al. (2023)). We present the concrete certified training approaches relevant to this work in Section 3. Generally, certified training methods are evaluated with regard to two metrics, *i.e.*, *clean* and *certified* accuracy where, given a test set, the former refers to the fraction of correctly classified inputs and the latter refers to the fraction of inputs for which the network is provably robust within $\mathcal{B}_\infty^\epsilon$.

## 2.3 MULTI-OBJECTIVE HYPERPARAMETER OPTIMISATION

**Formal definition.**  In hyperparameter optimisation, let $\mathcal{A}$ be an algorithm and $\mathbf{\Lambda}$ its configuration space, containing the hyperparameters and their ranges considered for optimisation. When $\mathcal{A}$ is run with a hyperparameter configuration $\lambda \in \mathbf{\Lambda}$, we denote it as $\mathcal{A}_\lambda$. Given a data distribution $D$ with training set $D_{\text{train}}$ and test set $D_{\text{test}}$, and $l$ performance metrics $\mathbf{m} = \{m_1, \ldots, m_l\}$, each metric evaluates the performance of $\mathcal{A}_\lambda$ trained on $D_{\text{train}}$ and tested on $D_{\text{test}}$. We assume w.o.l.g. that the optimisation goal is to maximise all metrics. We denote the metric values of configuration $\lambda$ as

$$\mathbf{m}(A_\lambda) = \big(m_1(A_\lambda),\, m_2(A_\lambda),\, \ldots,\, m_l(A_\lambda)\big). \qquad (5)$$

The optimisation may have *constraints* $c_1(A_\lambda), \ldots, c_k(A_\lambda)$. A configuration $\lambda$ satisfies constraint $c_i$ if, and only if, $c_i(\lambda) \geq 0$, and configurations satisfying all constraints are called *feasible*.

For two feasible configurations $\lambda_i, \lambda_j \in \mathbf{\Lambda}$, we say that $\lambda_i$ *Pareto dominates* $\lambda_j$ (i.e., $\mathbf{m}(A_{\lambda_i}) \succ \mathbf{m}(A_{\lambda_j})$) if

$$\forall k \in \{1, \ldots, l\} : m_k(A_{\lambda_i}) \geq m_k(A_{\lambda_j}) \quad \text{and} \quad \exists k \in \{1, \ldots, l\} : m_k(A_{\lambda_i}) > m_k(A_{\lambda_j}).$$

The optimisation goal is to identify the Pareto set of non-dominated feasible configurations $\Lambda^* \subseteq \mathbf{\Lambda}$, such that $\lambda \in \Lambda^*$ iff $\nexists \lambda' \in \mathbf{\Lambda}$ with $\mathbf{m}(A_\lambda) \prec \mathbf{m}(A_{\lambda'})$. The corresponding Pareto front of is denoted $\mathbf{M}^* = \{\mathbf{m}(A_\lambda) \mid \lambda \in \Lambda^*\}$.

Common metrics for assessing multi-objective optimisation include the *hypervolume*, defined as the Lebesgue measure of the dominated space between a reference point $r \in \mathbb{R}^l$ and an approximate Pareto front $\mathbf{M}$; we denote it as $\text{HV}(\mathbf{M}, r)$.

**Multi-objective Bayesian optimisation.**  Since many real-world problems involve multiple objectives, several approaches for multi-objective optimisation have been proposed, including evolutionary algorithms (Beume et al., 2007; Deb et al., 2002) and Bayesian optimisation (Daulton et al., 2020), the latter of which we adopt in this work. Bayesian optimisation is a surrogate-based approach that iteratively samples configurations $\lambda_1, \lambda_2 \ldots, \lambda_t$ and stores them in a dataset $\zeta$. This dataset is used to train *surrogate models* $S_1 : \hat{\Lambda} \to \mathbb{R}, S_2 : \hat{\Lambda} \to \mathbb{R}, \ldots, S_l : \hat{\Lambda} \to \mathbb{R}$, each approximating an objective $m_1, \ldots, m_l$. In addition to objective estimates, surrogates provide predictive uncertainty, typically expressed as a variance $\sigma^2$. Common choices for surrogate models include Gaussian processes (Rasmussen & Williams, 2006) and random forests (Breiman, 2001). An *acquisition function* balances exploration and exploitation, and selects the configuration with the highest acquisition value for evaluation. The dataset $\zeta$ is updated with this configuration, and the process continues until a given evaluation budget is exhausted.

In the multi-objective setting, the *expected hypervolume improvement* (EHVI) acquisition function is frequently used. Given a Pareto front $\mathbf{M}$ and a new configuration $\lambda \in \mathbf{\Lambda}$, the hypervolume improvement is defined as

$$\text{HVI}(\mathbf{M}, \lambda) = \big(\text{HV}(\mathbf{M} \cup \{\mathbf{m}(A_\lambda, D_{\text{train}}, D_{\text{test}})\}) - \text{HV}(\mathbf{M})\big) \cdot \mathbb{1}\,[\mathbf{c}(A_\lambda, D_{\text{train}}, D_{\text{test}})], \qquad (6)$$

*i.e.*, the additional hypervolume gained by adding $\lambda$ to the Pareto set. The EHVI is then given by $\text{EHVI}(\mathbf{M}, \lambda) = \mathbb{E}[\text{HVI}(\mathbf{M}, \lambda)]$

## 3 RELATED WORK

In the following, we give a brief overview of related work from the certified training and multi-objective hyperparameter optimisation literature.

**State-of-the-Art Certified Training Techniques.** As stated previously, state-of-the-art certified training relies on IBP to approximate the worst-case robust loss. This approach was first introduced by Gowal et al. (2019) but required gradually increasing $\epsilon$ to its final value over hundreds of *ramp-up* epochs to stabilise training. In addition, Gowal et al. introduced a trade-off parameter $\kappa$ that is decreased from 1 to 0 during ramp-up, weighing clean and verified loss: $\kappa \cdot \mathcal{L}(f_\theta(\mathbf{x}), y) + (1 - \kappa) \cdot \mathcal{L}_{\text{ver}}(f_\theta(\mathbf{x}), y)$. Prior to certified training, the network may be initialized with several *warm-up* epochs using the clean loss. Zhang et al. (2020) propose to combine IBP and CROWN (Zhang et al., 2018) bounds in *CROWN-IBP* to compute $L_{\text{ver}}$. Here, CROWN relaxations are used to bound the final output based on IBP bounds of intermediate layers. Furthermore, a transition is made from CROWN-IBP to IBP bounds during ramp-up, using an additional trade-off parameter $\beta$. Xu et al. (2020) further reduce the complexity of CROWN-IBP through *loss fusion*, a technique that enables direct computation of the verified loss without requiring logit differences. Shi et al. (2021) suggest the use of BatchNorm layers (Ioffe & Szegedy, 2015) and introduce specialised initialisation and regularisation techniques resulting in shorter ramp-up schedules and better performance. More recently, a line of research emerged that combines certified and adversarial losses. Müller et al. (2023) compute an unsound verified loss called *SABR* by propagating a smaller subset of the input region with edge length $\tau \cdot \epsilon$ using IBP. The centers of the hyper-box are identified using PGD. Additionally, *ReLU shrinking* is used to reduce the magnitude of IBP bounds by multiplying them with a constant $c < 1$ before each activation, thereby gradually increasing focus on adversarial loss. De Palma et al. (2024b) show that loss functions conceptually similar to SABR can be obtained by considering convex combinations of $\mathcal{L}_{\text{ver}}$ and $\mathcal{L}_{\text{adv}}$ weighed by $\alpha$. Among those, the *MTL-IBP* loss is defined as $\alpha \cdot \mathcal{L}_{\text{ver}} + (1 - \alpha) \cdot \mathcal{L}_{\text{adv}}$. In addition, an effect similar to ReLU shrinking is achieved by carrying out adversarial attacks over a larger perturbation radius.

**Evaluation of Certified Training.** To assess certified accuracy of trained models, related work employed state-of-the-art complete verification systems *Oval* (De Palma et al., 2024a) or *MN-BaB* (Ferrari et al., 2022). In addition, the tuning of parameters including the learning rate, the number of warm- and ramp-up epochs and trade-off parameters, such as $\kappa$ or $\alpha$, is crucial for achieving state-of-the-art performance. Until now, researchers have mostly relied on tuning parameters manually to obtain a single configuration that compares favourably to the current state of the art. Recently, Mao et al. (2025) proposed *CTBench*, a novel benchmark for certified training, with the goal of ensuring a fair comparison between methods by employing grid search over separately designed hyperparameter spaces per benchmark. Nevertheless, the results presented in CTBench were obtained by tuning to one specific trade-off that often favoured certified accuracy and, thus, came at the expense of markedly reduced clean accuracy on some benchmarks as we show later in Section 5.

**Multi-Objective Hyperparameter Optimisation.** Multi-objective optimisation was deployed previously in multiple AutoML scenarios. For example, Dooley et al. (2023) performed joint hyperparameter optimisation and neural architecture search of CNNs to train networks which are not only accurate but also unbiased. Hennig & Lindauer (2025) used multi-objective hyperparameter optimisation to find optimal shift neural networks that balance energy efficiency and accuracy. The popular YAHPO (Pfisterer et al., 2022) benchmark offers several multi-objective hyperparameter optimisation benchmarks for tabular machine learning. The benchmarks balance between different objectives, including accuracy, memory usage and interpretability.

## 4 PARETO-FRONT DISCOVERY OF CERTIFIED TRAINING METHODS

In the following, we present our novel method for the discovery of a Pareto-optimal set of hyperparameter configurations for certified training. With this, we address multiple open problems in the literature. First and foremost, we present a fully-automated pipeline to obtain optimal configurations for state-of-the-art methods. This renders labour-intensive manual hyperparameter tuning unnecessary, thereby making the process more accessible to non-experts and more efficient for experts. Further, it offers a principled approach to finding high-performance configurations that might reveal new trade-offs between clean and certified accuracy that could not be found through manual tuning. In addition, the Pareto fronts enable a more nuanced comparison of certified training techniques, *e.g.*, they may uncover that one method yields more favourable trade-offs at a certain level of clean accuracy than another.

**Search space design.** Since certified training depends on several hyperparameters whose influence on performance is not known *a priori*, we opted to include all relevant hyperparameters in our search space. These include general hyperparameters of deep learning pipelines, such as the learning rate, epochs at which the learning rate is decayed and the optimiser used to find best performing parameters with regard to Equation 3 (*e.g.*, Adam (Kingma & Ba, 2015) or RAdam (Liu et al., 2020)). Furthermore, we adapt $\ell_1$ regularisation, since it has proven beneficial for certified training, and optimise its weight-parameter. Regarding techniques specific to certified training, we optimise for the weight of the regularisation proposed by Shi et al. (2021), which is employed in all state-of-the-art methods (see, *e.g.*, De Palma et al. (2024b); Müller et al. (2023)). Furthermore, we search for an optimal number of warm-up and ramp-up epochs. It may also be beneficial to train with a larger perturbation radius than used for evaluation (see, *e.g.*, De Palma et al. (2024b); Gowal et al. (2019)); hence, we optimise a parameter that scales the $\epsilon$ value used in training.

Moreover, we search for optimal method-specific trade-off parameters $\tau$ for SABR- and $\alpha$ for MTL-IBP-based training. Regarding $\kappa$, we optimise two parameters $\kappa_{\text{start}} \geq \kappa_{\text{end}}$ and transition from $\kappa_{\text{start}}$ to $\kappa_{\text{end}}$ during the ramp-up phase. We handle the $\beta$ parameter in CROWN-IBP analogously. For SABR and MTL-IBP, we additionally optimise the number of PGD steps and their step size. To keep training cost tractable, we do not restart PGD multiple times per batch, as done by Mao et al. (2025); a choice consistent with several prior studies (see, *e.g.*, De Palma et al. (2024b); Madry et al. (2018)). Lastly, we tune the $\epsilon$-radius over which the PGD attack is carried out.

Overall, we constructed the search space to include all plausible parameter choices, rather than restricting it to those previously shown to be successful in the literature. If those choices were indeed optimal, we rely on the optimiser to discover them during search. For example, we included $\kappa$ and $\beta$ as optimisable parameters, while related work has deemed those transitions unnecessary, and we allow up to five warm-up epochs, while related work employed at most one (see, *e.g.*, (Mao et al., 2025; Shi et al., 2021)). With this, we hope to uncover previously unexplored configurations that yield better trade-offs than prior work. We present the complete search space in Appendix B.7.

**Optimisation metrics.** As outlined previously, metrics of interest for certified training are clean and certified accuracy. While evaluating clean accuracy is cheap, evaluating certified accuracy with complete verification systems for each configuration is computationally infeasible. Thus, we optimise for an under-approximation of the true certified accuracy by employing the incomplete verification methods IBP, CROWN-IBP and CROWN, running computationally more demanding methods only when cheaper methods could not provide a result.

**Search strategy.** Since hyperparameters are often inter-dependent (Moosbauer et al., 2021), we jointly optimise all hyperparameters within the previously defined search space. To identify configurations that optimally balance certified and natural accuracy, we employ multi-objective optimisation. However, we do not want to focus on regions of the Pareto front exhibiting high natural accuracy with extremely low certified accuracy, or *vice versa*, which can be obtained, *e.g.*, by tuning SABR and MTL-IBP to reduce to adversarial training. Therefore, we constrain the optimisation to an area of interest to avoid spending expensive resources on uninteresting configurations.

Multi-fidelity approaches are commonly used in hyperparameter optimisation to improve efficiency (Eggensperger et al., 2021; Dooley et al., 2023). They first assess many configurations at low fidelities (*e.g.*, fewer training epochs) and reserve high-fidelity evaluations for promising candidates. In certified training, however, the ramp-up phase prevents meaningful comparison before training completes, so we leave this extension for future work.

For the reasons mentioned above, we employ multi-objective Bayesian optimisation with a Gaussian process surrogate and an EHVI acquisition function accommodating constraints. As the optimisation objectives are independent from each other, we model them using distinct Gaussian processes. To avoid undesirable outcomes, such as becoming trapped in local optima or over-exploration of specific parts of the Pareto front, we execute the optimisation with three pseudo-random seeds. We then combine the Pareto fronts discovered by those three runs to create a single, unified Pareto front.

**Complete verification.** To obtain the final Pareto front, we assess the performance of all Pareto-optimal configurations found with regard to incomplete verification using a state-of-the-art com-

Table 1: Comparison of the results reported from the literature to the results achieved by using our novel optimisation procedure. For each result from the literature, we selected a configuration from the Pareto front that achieves similar or better performance. Boldface marks results surpassing prior work; underlined values indicate similar performance ($\pm 0.5$). Our method typically yields configurations with higher clean accuracy and, in many cases, improved certified accuracy.

| Dataset | $\epsilon$ | Method | Source | Clean Acc. [%] (Lit.) | Cert. Acc. [%] (Lit.) | Clean Acc. [%] (ours) | Cert. Acc. [%] (ours) |
|---|---|---|---|---|---|---|---|
| CIFAR-10 | $\frac{2}{255}$ | MTL-IBP | De Palma et al. (2024b) | 80.11 | 63.24 | 79.97 | **63.99** |
| | | MTL-IBP | Mao et al. (2025) | 78.82 | 64.41 | **79.87** | 64.54 |
| | | SABR | Müller et al. (2023) | 79.24 | 62.84 | **81.95** | **64.11** |
| | | SABR | Mao et al. (2025) | 77.86 | 63.61 | **80.15** | **64.44** |
| | | IBP | Shi et al. (2021) | 66.84 | 52.85 | **71.39** | **55.54** |
| | | IBP | Mao et al. (2025) | 67.49 | 55.99 | **69.37** | 55.62 |
| | | CROWN-IBP | Zhang et al. (2020) | 71.52 | 53.97 | **77.44** | **59.25** |
| | | CROWN-IBP | Mao et al. (2025) | 67.60 | 57.11 | **75.70** | **61.39** |
| | $\frac{8}{255}$ | MTL-IBP | De Palma et al. (2024b) | 53.35 | **35.44** | **55.25** | 34.49 |
| | | MTL-IBP | Mao et al. (2025) | 54.28 | 35.41 | 54.18 | 35.27 |
| | | SABR | Müller et al. (2023) | 52.38 | 35.13 | **54.93** | 34.96 |
| | | SABR | Mao et al. (2025) | 52.71 | **35.34** | **56.06** | 34.26 |
| | | IBP | Shi et al. (2021) | 48.94 | 34.97 | **52.62** | 35.09 |
| | | IBP | Mao et al. (2025) | 48.51 | 35.28 | **51.02** | 35.35 |
| | | CROWN-IBP | Zhang et al. (2020) | 46.29 | 33.38 | **55.11** | 33.77 |
| | | CROWN-IBP | Mao et al. (2025) | 48.25 | 32.59 | **52.47** | **34.31** |
| Tiny ImageNet | $\frac{1}{255}$ | MTL-IBP | De Palma et al. (2024b) | 37.56 | 26.09 | **39.80** | **30.45** |
| | | MTL-IBP | Mao et al. (2025) | 35.97 | 27.73 | **39.75** | **30.67** |
| | | SABR | Müller et al. (2023) | 28.85 | 20.46 | **40.61** | **28.86** |
| | | SABR | Mao et al. (2025) | 30.58 | 20.96 | **42.10** | **26.38** |
| | | IBP | Shi et al. (2021) | 25.92 | 17.87 | **34.24** | **20.03** |
| | | IBP | Mao et al. (2025) | 26.77 | 19.82 | **32.12** | **21.53** |
| | | CROWN-IBP | Xu et al. (2021) | 25.62 | 17.93 | **32.38** | **20.72** |
| | | CROWN-IBP | Mao et al. (2025) | 28.44 | 22.14 | **30.82** | 22.20 |

plete verification system. However, the front may include several configurations with negligible performance differences, for which complete verification would incur unnecessary costs. Therefore, in cases where more than 5 configurations are part of the Pareto front, we employ single-linkage clustering (Sibson, 1973), which starts by assigning each configuration to its own cluster and then iteratively merges close clusters whenever the Euclidean distance between the metrics of configurations from two clusters is less than $d_{\min}$. We evaluate one random configuration for each cluster and construct the final Pareto front using the certified accuracies obtained through complete verification.

## 5  EMPIRICAL EVALUATION

In the following, we describe the evaluation of our approach on standard benchmarks from the certified training literature.

**Setup of experiments.** For our experiments, we employed the certified training implementations of CTRAIN (Kaulen & Hoos, 2025), focusing on IBP, CROWN-IBP, SABR and MTL-IBP as the methods under investigation. With this, we aimed to include current state-of-the-art methods as well as seminal advancements from the field. For the hyperparameter optimiser, we used BoTorch (Balandat et al., 2020) within the Optuna package (Akiba et al., 2019), which provides an implementation of our chosen optimisation algorithm. Based on preliminary experiments, we set the evaluation budget for each optimisation run to 100 trials, resulting in 300 trials per benchmark. For complete verification, we used the state-of-the-art (Brix et al., 2024) verification system $\alpha\beta$-CROWN (Wang et al., 2021; Xu et al., 2021) with a cutoff of 1 000 seconds in wall-clock time. For comparability with related work, we followed the seemingly common practice in the certified training community of tuning hyperparameters on the test set (see, *e.g.*, Mao et al. (2025); Shi et al. (2021)).

We considered the *CNN7* architecture of Shi et al. (2021), the *de facto* standard architecture for evaluating certified training methods (see, *e.g.*, De Palma et al. (2024b); Müller et al. (2023)). We present results on CIFAR-10 (Dosovitskiy et al., 2021) for $\epsilon$-radii $\frac{2}{255}$ and $\frac{8}{255}$ and on Tiny ImageNet (Le & Yang, 2015) for $\epsilon = \frac{1}{255}$, following general evaluation protocol of De Palma et al. (2024b)

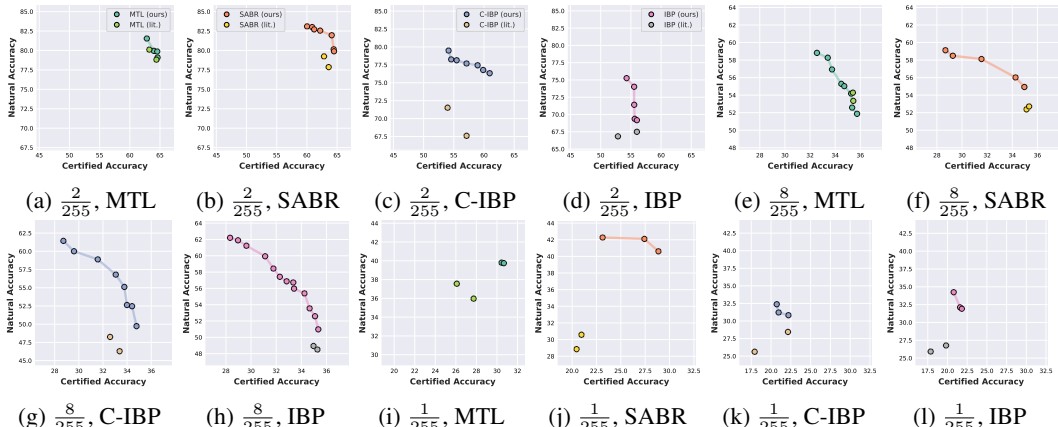

(a) $\frac{2}{255}$, MTL  (b) $\frac{2}{255}$, SABR  (c) $\frac{2}{255}$, C-IBP  (d) $\frac{2}{255}$, IBP  (e) $\frac{8}{255}$, MTL  (f) $\frac{8}{255}$, SABR

(g) $\frac{8}{255}$, C-IBP  (h) $\frac{8}{255}$, IBP  (i) $\frac{1}{255}$, MTL  (j) $\frac{1}{255}$, SABR  (k) $\frac{1}{255}$, C-IBP  (l) $\frac{1}{255}$, IBP

Figure 1: Results for CIFAR-10 for $\epsilon = \frac{2}{255}$ are shown in (a)-(d), for $\epsilon = \frac{8}{255}$ in (e)-(h), and for Tiny ImageNet for $\epsilon = \frac{1}{255}$ in (i)-(l). We compare Pareto fronts obtained using our method to results given in the original publications and the recent CTBench benchmark (Mao et al., 2025).

(see Appendix B). We set $d_{\min} = 0.05$ to filter redundant configurations and restrict the optimisation process to configurations meeting minimum certified and natural accuracies of $40\%$ and $60\%$ for CIFAR-10 ($\epsilon = \frac{2}{255}$), $25\%$ and $40\%$ ($\epsilon = \frac{8}{255}$), and $15\%$ and $20\%$ for Tiny ImageNet; these limits were chosen based on the results reported in the original publications. Furthermore, we chose to run CROWN-IBP without loss fusion on CIFAR-10, since this resulted in generally superior performance. Additional results on MNIST (LeCun, 1998) and on different architectures, including a wider *CNN7* used by Mao et al. (2024), are provided in Appendix C.

**Comparison to previously-known results.** We begin by examining the configurations found using our optimisation procedure to previously-known results. Table 1 compares the results achieved by our method to those from the literature, including the original publications of each method and the recent CTBench benchmark (Mao et al., 2025). As previous studies reported only a single configuration, we selected Pareto-optimal configurations that either dominate or match them. In nearly all scenarios, the performance of configurations equals or surpasses prior results.

On CIFAR-10 with $\epsilon = \frac{2}{255}$, SABR achieves a gain of more than $1\%$ in terms of clean and certified accuracy, surpassing prior results and setting a new state of the art. Furthermore, our results demonstrate that MTL-IBP can achieve strong certified and clean performance at the same time. For CROWN-IBP and IBP, we found that these older methods remain competitive, with CROWN-IBP achieving nearly a $6\%$ improvement in clean accuracy over best results from the literature.

While for $\epsilon = \frac{8}{255}$, our optimisation did not outperform previously known results regarding certified accuracy, it often finds configurations with comparable certified but higher natural accuracy. Mao et al. (2025) suggest that all investigated methods converge to the same certified accuracy at this larger perturbation radius. We validate this result but show that the performance differences regarding clean accuracy are much less pronounced.

For TinyImageNet, we obtain new state-of-the-art results that substantially surpass prior work, with MTL-IBP achieving an improvement of about $2\%$ in terms of clean and certified accuracy. We further demonstrate that SABR can achieve comparable results.

**Comparison between methods.** The Pareto fronts obtained from our novel method allow for a more nuanced and multi-faceted assessment of the current state of the art in certified training. Instead of comparing single configurations, it is now possible to evaluate the quality of feasible solutions across the entire trade-off space. For this, we combined all configurations found by our method into one single Pareto front per dataset and perturbation radius and analysed which methods contribute to this combined front. We show the Pareto fronts of all methods per benchmark in Figure 2.

Regarding the CIFAR-10 dataset with $\epsilon = \frac{2}{255}$, the combined Pareto set consists of networks trained with MTL-IBP and SABR. Our analysis reveals that SABR generally achieves the highest clean ac-

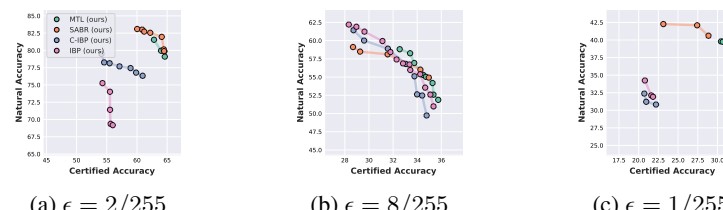

(a) $\epsilon = 2/255$        (b) $\epsilon = 8/255$        (c) $\epsilon = 1/255$

Figure 2: Comparison of Pareto fronts from our method on CIFAR-10 with (a) $\epsilon = \frac{2}{255}$, (b) $\epsilon = \frac{8}{255}$ and Tiny ImageNet with (c) $\epsilon = \frac{1}{255}$. The fronts enable a nuanced assessment, showing, *e.g.*, that IBP is state of the art in (b) when prioritising natural accuracy.

curacies while maintaining strong certified robustness. However, MTL-IBP can still achieve similar certifiable guarantees once clean accuracy decreases. For the higher perturbation radius of $\epsilon = \frac{8}{255}$, we found that traditional IBP training contributes to the combined front alongside MTL-IBP and SABR. More specifically, networks trained with IBP exhibit the strongest certifiable guarantees for higher clean accuracies, while SABR and MTL-IBP achieve better trade-offs for higher certified accuracies. This shows that IBP is a state-of-the-art method when higher natural accuracies are desired. Lastly, on Tiny ImageNet, the Pareto front includes networks trained using SABR and MTL-IBP. Here, SABR excels at increased natural accuracies, while MTL-IBP performs best when higher certified accuracies are desired.

# 6 CONCLUSIONS AND FUTURE WORK

In this work, we have proposed a novel method for the fully-automated hyperparameter optimisation of certified training techniques. Using this method, we tackle several open challenges in certified training. Firstly, until now, hyperparameter tuning required extensive domain knowledge and was not accessible to non-experts. Our automated optimisation pipeline removes this barrier by systematically exploring the hyperparameter space and identifying configurations that achieve favourable trade-offs between clean and certified accuracy. Secondly, prior evaluations of certified training methods typically focused on single configurations, limiting insight into the overall performance landscape. By constructing Pareto fronts of configurations, our method enables a more comprehensive assessment of the trade-offs that can be achieved, highlighting which certified training techniques perform well consistently. Lastly, using our approach, we have demonstrated that there exist more optimal trade-offs than previously known for several popular certified training methods including MTL-IBP and SABR, thereby establishing a new state of the art in certified training.

To achieve this, we employed techniques from constrained multi-objective hyperparameter optimisation in a novel tuning scheme that search for optimal trade-offs within an expert-designed search space. For its design, we ensured to include all potentially sensible hyperparameter choices to enable the discovery of previously unexplored configurations. Furthermore, we constrained the optimisation process to exclusively explore promising regions of the search space, in order to prevent a focus on trivial configurations that reduce to adversarial or standard training. Lastly, since complete verification for every configuration is computationally infeasible, our optimisation relies on a proxy metric. We showed that incomplete verification enables efficient assessment of certifiability, allowing the selection of configurations that also perform well under complete verification.

For future work, we suggest investigating how multi-fidelity optimisation and meta-learning techniques for Bayesian optimisation (see, *e.g.*, Dooley et al. (2023); Feurer et al. (2018)) could be adapted for certified training to further improve efficiency.

Overall, we believe that evaluation of certified training techniques should focus on Pareto front analysis rather than results for single hyperparameter configurations. By providing a method to effectively approximate the Pareto front, our work establishes a foundation for a more nuanced evaluation and calibration of certified training techniques.

## 7 ETHICS STATEMENT

Our paper aims to improve the performance of certified training methods, providing a full Pareto front of well-performing configurations with different accuracy-robustness trade-offs.. As certified training methods are used to obtain provably safe neural networks, we see no negative ethical contributions of our work. Further, our Pareto front analysis enables a nuanced assessment of the performance of certified training techniques, thereby facilitating their responsible and informed application in practice. However, while we demonstrated the effectiveness of our method across several commonly used vision datasets, this does not guarantee its effectiveness on different benchmarks, data domains or threat models.

## 8 REPRODUCIBILITY STATEMENT

Our code is available in anonymous GitHub repository: `https://anonymous.4open.science/r/investigating_certified_training_trade_offs-0584`. In our experiments, we used popular open-source datasets which can be downloaded and preprocessed via CTRAIN (Kaulen & Hoos, 2025). We provide additional information on the setup of our experiments in Appendix B, including hardware details, software versions, neural network architectures used and detailed configuration spaces.

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

# A    HYPERPARAMETER IMPORTANCE ANALYSIS

In the following, we aim to uncover reasons to why the discovered hyperparameter configurations perform better than previously known configurations. To this end, we use fANOVA (Hutter et al., 2014a) to identify the hyperparameters that were most influential during the optimisation procedure. fANOVA quantifies the importance of a hyperparameter (or set of hyperparameters) as the fraction of the variance in the predicted performance that can be attributed to it. Intuitively, changing an important hyperparameter is expected to have a large effect on performance. To estimate this, fANOVA fits a random forest (Breiman, 2001) as a surrogate model and computes the marginal effect of a hyperparameter by integrating over all possible values of the other hyperparameters.

In Figures 3, 4, 5 and 6 we display the five most important hyperparameters for each objective (*i.e.*, clean and certified accuracy) on the CIFAR-10 dataset for $\epsilon \in \{\frac{2}{255}, \frac{8}{255}\}$ along with the parameter values of the configurations in the Pareto set for the training methods IBP, CROWN-IBP, SABR and MTL-IBP respectively. Further, the plots include the achieved clean accuracy or, respectively, the certified accuracy obtained through incomplete verification.

**IBP.**    For most investigated scenarios, our analysis reveals that IBP yields stronger trade-offs when more time during training is spent on optimising for clean cross-entropy loss. This is exemplified in $\kappa_{\text{start}}$ and $\kappa_{\text{end}}$ being highly important parameters across all scenarios, with higher $\kappa$ values than used previously. Interestingly, when $\epsilon = \frac{8}{255}$, scaling the training $\epsilon$ has high influence on both clean and certified accuracy and is even the most important hyperparameter for achieving strong clean accuracy.

**CROWN-IBP.**    Regarding CROWN-IBP, we observe a similar trend where $\kappa$ parameters play an important role in achieving strong performance across all scenarios to trade-off clean and certified accuracy. Again, the factor by which the $\epsilon$ value is scaled during training plays an important role as well. Interestingly, $\beta_{\text{end}}$ is an important parameter to tune certified accuracy with different optimal values between both $\epsilon$ values. For $\epsilon = \frac{2}{255}$, higher $\beta$ values, *i.e.* a higher focus on CROWN-IBP bounds, yield better performance while for $\epsilon = \frac{8}{255}$ it is crucial that $\beta \approx 0$ at the end of the ramp-up phase to achieve strong certifiable guarantees.

**SABR.**    When investigating the results for SABR, it becomes apparent that the subselection ratio $\tau$ is extremely effective at governing the trade-off between certified and clean accuracy, being a highly important parameter across all scenarios. Further, parameters of the employed attack are also highly important, such as the number of optimisation steps or the scaling factor of the $\epsilon$ applied during the attack. For the latter, interestingly, higher values result in higher certified accuracies when $\epsilon = \frac{2}{255}$, but when $\epsilon = \frac{8}{255}$, the opposite is the case. Most importantly, our analysis reveals a simple and intuitive explanation to why SABR achieves stronger natural accuracies than the previous state of the art when $\epsilon = \frac{2}{255}$. Here, a highly important parameter is the choice of the optimiser which is always set to RAdam (Liu et al., 2020) for all configurations in the Pareto set. This optimiser seems to be able to achieve substantially better trade-offs in this scenario than were previously known.

**MTL-IBP.**    Lastly, we focus on the hyperparameter configurations of MTL-IBP. The method-inherent trade-off parameter $\alpha$ is very important across all scenarios, effectively steering the trade-off between natural and certified accuracy. Interestingly, our method found several configurations that achieve, both, strong certified and natural accuracies when $\epsilon = \frac{2}{255}$, which could be traced back to a higher number of warm-up epochs employed in our configurations than in related work. Interestingly, when $\epsilon = \frac{8}{255}$, the weight of the regulariser proposed by Shi et al. (2021) seems to have major impact on the achievable natural accuracy. Regarding certified accuracy on that benchmark, more PGD steps correspond to higher certified accuracies, indicating that a better approximation of the adversarial loss is crucial in this case.

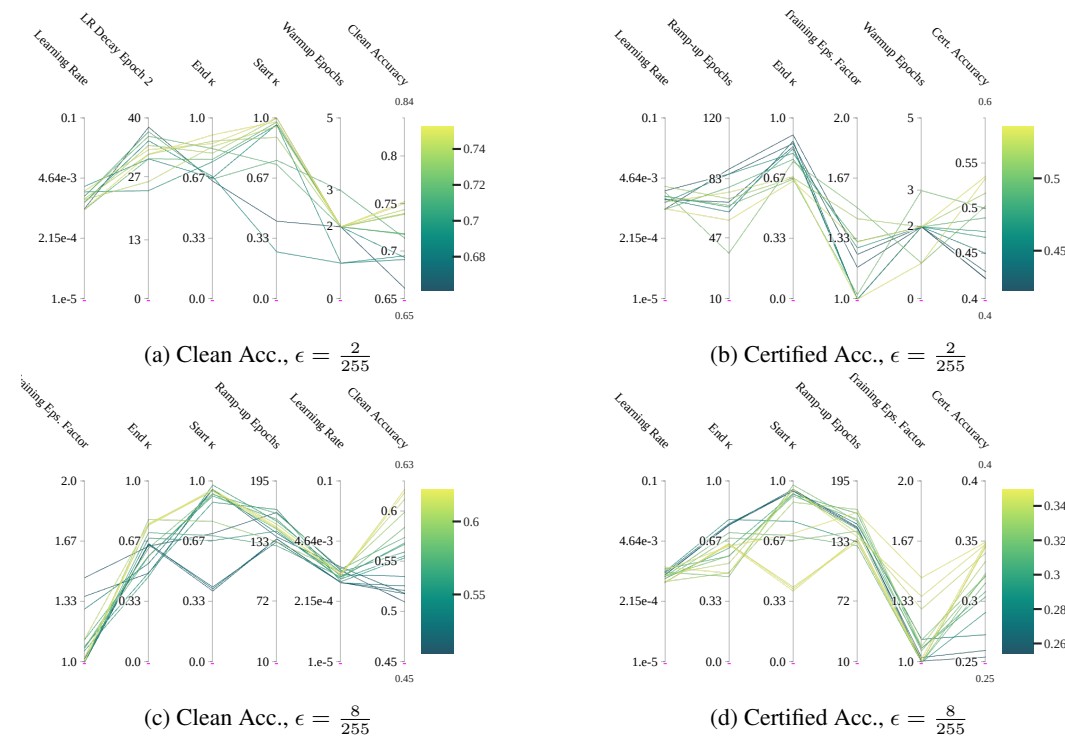

Figure 3: Parallel coordinates plot for the hyperparameter optimisation of IBP on CIFAR-10. In each plot, we show the five most important parameters for one of the two objectives along with the parameter values of configurations in the Pareto set.

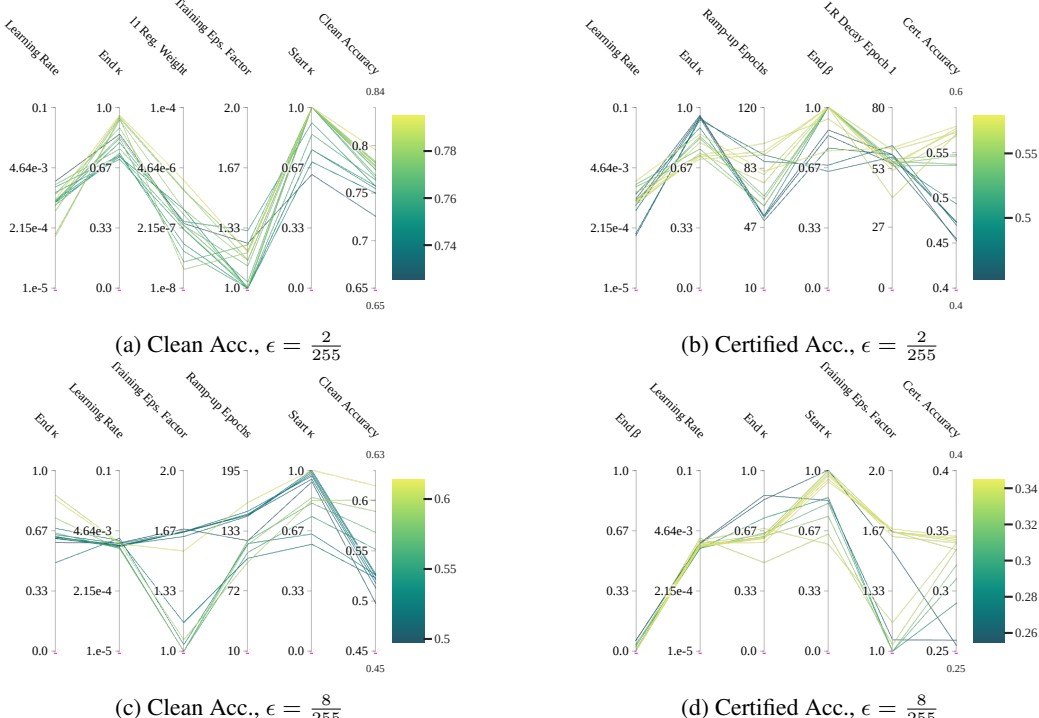

Figure 4: Parallel coordinates plot for the hyperparameter optimisation of CROWN-IBP on CIFAR-10. In each plot, we show the five most important parameters for one of the two objectives along with the parameter values of configurations in the Pareto set.

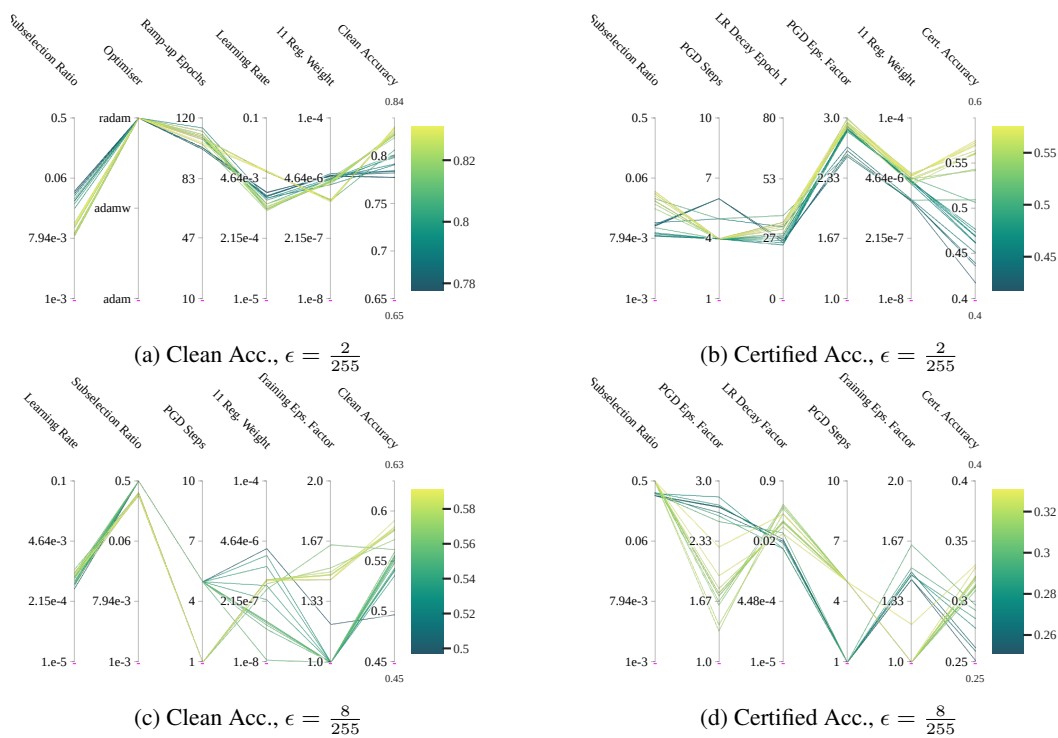

Figure 5: Parallel coordinates plot for the hyperparameter optimisation of SABR on CIFAR-10. In each plot, we show the five most important parameters for one of the two objectives along with the parameter values of configurations in the Pareto set.

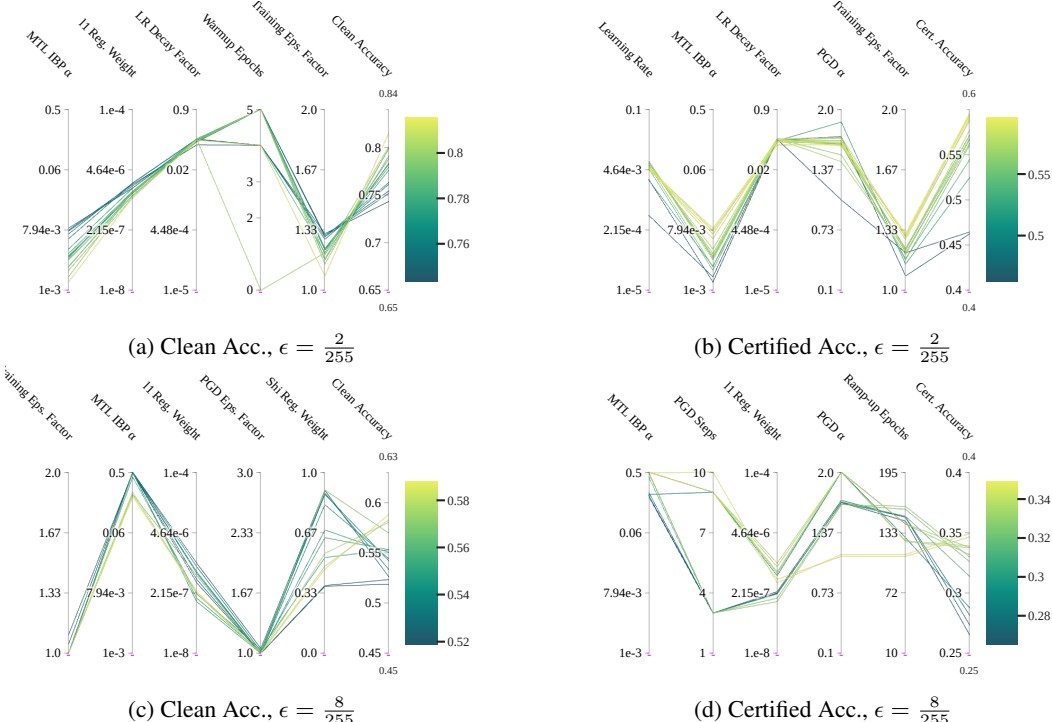

Figure 6: Parallel coordinates plot for the hyperparameter optimisation of MTL-IBP on CIFAR-10. In each plot, we show the five most important parameters for one of the two objectives along with the parameter values of configurations in the Pareto set.

## B    Additional Details on the Setup of the Experiments

### B.1    Hardware Details

Our experiments were conducted on two compute clusters. For running our method on CIFAR10, as well as calculating certified accuracy using complete verification for all datasets except MNIST we used a cluster in which each node is equipped with two Intel Xeon Platinum 8480+ with 210MB of L3 cache, four Nvidia H100 SXM GPUs, and 2TB of RAM running Rocky Linux 9. Each optimisation and verification experiment utilised 14 CPU cores, 220GB of RAM and one GPU. For running our method on TinyImageNet and MNIST, as well as verifying the obtained Pareto fronts on MNIST, we used a cluster in which each node is equipped with two Intel Xeon Platinum 8468 with 210MB of L3 cache, four Nvidia H100 NVL GPUs, 512GB of RAM running Rocky Linux 9. Here, each experiment utilised 24 CPU cores, 120GB of RAM and one GPU.

### B.2    Datasets

For our experiments, we employed several well-known datasets that have been used regularly within the certified training community. First, we used CIFAR10 (Krizhevsky et al., 2009) containing RGB images (*i.e.*, three channels) of size $32 \times 32$ pixels associated to 10 classes (such as airplane, frog, ...). The dataset includes $50\,000$ training samples and $10\,000$ test samples. TinyImageNet (Le & Yang, 2015) is a subsampled version of the ImageNet dataset including $100\,000$ training samples and $10\,000$ test samples restricted to 200 classes. The resolution of each image is $64 \times 64$ with three channels. In additional experiments, we employed the MNIST dataset (LeCun, 1998) which contains grayscale images of size $28 \times 28$ with $60\,000$ training samples and $10\,000$ test samples. In line with previous work (see, *e.g.*, Mao et al. (2025); De Palma et al. (2024b); Müller et al. (2023); Shi et al. (2021); Xu et al. (2020)), we normalised all datasets and used data augmentations when training on CIFAR-10 and TinyImageNet. More specifically, we augmented CIFAR-10 and TinyImageNet with random horizontal flips and random cropping to $32 \times 32$ pixels after 2-pixel padding for CIFAR-10, and to $64 \times 64$ pixels after 4-pixel padding for TinyImageNet. Lastly, we trained on the corresponding train sets and report clean and certified accuracy on the test split (in line with, *e.g.*, Mao et al. (2025); De Palma et al. (2024b); Müller et al. (2023); Shi et al. (2021); Xu et al. (2020)).

### B.3    Architectures

We use the CNN7 architecture from Shi et al. (2021) across all datasets and $\epsilon$ radii, which is the *de facto* standard architecture to evaluate certified training methods on (see, *e.g.*, Mao et al. (2025); De Palma et al. (2024b); Mao et al. (2024); Müller et al. (2023)). This architecture employs Batch-Norm layers (Ioffe & Szegedy, 2015) before every ReLU activation which improve the performance of certified training methods by reducing an imbalance between active and inactive neurons (Shi et al., 2021). To further evaluate the consistency of our tuning method and to investigate findings by Mao et al. (2024) regarding the influence of architecture on certified training, we included wide and narrow variants of CNN7 as defined by Mao et al. (2024) in additional experiments. We also show the performance on deeper and shallower versions of CNN7, named CNN9 and CNN5, respectively following Mao et al. (2024). The architectures are illustrated in Table 2.

### B.4    Experimental Setup

Generally, we mostly followed the experimental setup of De Palma et al. (2024b) for our hyperparameter optimisation. In the following, we give a detailed description how we ran the respective certified training methods during our novel optimisation method.

**Initialisation.**    Before the start of the training procedure, the network is initialised using the technique proposed by Shi et al. (2021) that relies on a low-variance Gaussian distribution to prevent the *explosion* of IBP bounds during early training stages. This initialisation has been used in all recent works (see, *e.g.*, Mao et al. (2025); De Palma et al. (2024b); Müller et al. (2023)) and was found to be generally beneficial to performance.

Table 2: Model architectures of the CNN7, CNN5 and CNN9 architectures as defined by Shi et al. (2021) and Mao et al. (2024). For CNN7, we provide the number of filters for narrow and wide variants in the order of (Narrow CNN7— CNN7 — Wide CNN7)

(a) (Narrow, Wide) CNN7

| Convolutional: (32—64—128) filters of size $3 \times 3$, stride 1, padding 1
Batch normalisation
ReLU activation |
| --- |
| Convolutional: (32—64—128) filters of size $3 \times 3$, stride 1, padding 1
Batch normalisation
ReLU activation |
| Convolutional: (64—128—256) filters of size $3 \times 3$, stride 2, padding 1
Batch normalisation
ReLU activation |
| $2 \times$ Convolutional: (64—128—256) filters of size $3 \times 3$, stride 1, padding 1
Batch normalisation
ReLU activation |
| Linear: 512 neurons
Batch normalisation
ReLU activation |
| Linear: no. of classes in dataset |

(b) CNN5

| Convolutional: 64 filters of size $3 \times 3$, stride 1, padding 1
Batch normalisation
ReLU activation |
| --- |
| Convolutional: 64 filters of size $3 \times 3$, stride 2, padding 1
Batch normalisation
ReLU activation |
| Convolutional: 128 filters of size $3 \times 3$, stride 2, padding 1
Batch normalisation
ReLU activation |
| Linear: 512 neurons
Batch normalisation
ReLU activation |
| Linear: no. of classes in dataset |

(c) CNN9

| $2 \times$ Convolutional: 64 filters of size $3 \times 3$, stride 1, padding 1
Batch normalisation
ReLU activation |
| --- |
| Convolutional: 128 filters of size $3 \times 3$, stride 2, padding 1
Batch normalisation
ReLU activation |
| $4 \times$ Convolutional: 128 filters of size $3 \times 3$, stride 1, padding 1
Batch normalisation
ReLU activation |
| Linear: 512 neurons
Batch normalisation
ReLU activation |
| Linear: no. of classes in dataset |

**Training schedule.** At the beginning of training, we employ a defined number of *warm-up* epochs where the standard cross-entropy loss is used. After that, the perturbation radius $\epsilon$ used for the calculation of (CROWN-)IBP bounds and during the PGD attack is gradually increased starting at 0 until it reaches its final value $\epsilon_{\text{train}}$ over a defined number of *ramp-up* epochs. To anneal to the final $\epsilon$ value, early works employed a linear schedule (Gowal et al., 2019; Zhang et al., 2020), but more recently a smoothed schedule was found to yield better results (see, *e.g.*, Mao et al. (2025); De Palma et al. (2024b); Müller et al. (2023); Xu et al. (2020)). Here, $\epsilon$ is increased exponentially for the first 25% of ramp-up epochs and linearly thereafter. This leads to smaller $\epsilon$ values during the beginning of the training process, which contributes to training stability. Notice, that the $\epsilon$ radius used during training does not need to match the $\epsilon$ value used for evaluation. In some cases, training with a larger $\epsilon$ radius than that used for evaluation has been shown to be beneficial (see, e.g., Shi et al. (2021); Gowal et al. (2019)). For IBP and CROWN-IBP training, we chose to include additional parameters that are annealed during the ramp-up phase. Both methods employ a $\kappa$ parameter (Zhang et al., 2020; Gowal et al., 2019) which weighs certified with clean loss, *i.e.*, $\kappa \cdot \mathcal{L}(f_\theta(\mathbf{x}), y) + (1 - \kappa) \cdot \mathcal{L}_{\text{ver}}(f_\theta(\mathbf{x}), y))$. During ramp-up $\kappa$ smoothly transitions from $\kappa_{\text{start}}$ to $\kappa_{\text{end}}$, where $\kappa_{\text{start}} \geq \kappa_{\text{end}}$. Analogously, for CROWN-IBP we included the $\beta$ parameter (Zhang et al., 2020) that additionally weighs verified losses obtained through CROWN-IBP and IBP to calculate the final verified loss used, *i.e.*, $\mathcal{L}_{\text{ver}}(\mathbf{x}, y) = \beta \cdot \mathcal{L}_{\text{CROWN-IBP}}(\mathbf{x}, y) + (1 - \beta) \cdot \mathcal{L}_{\text{IBP}}(\mathbf{x}, y)$. This parameter transitions from $\beta_{\text{start}}$ to $\beta_{\text{end}}$ with $\beta_{\text{start}} \geq \beta_{\text{end}}$. This way, tighter CROWN-IBP bounds are only employed to stabilise the beginning of the training process which may result in superior performance to using CROWN-IBP bounds throughout (Zhang et al., 2020). However, it is important to notice that by setting $\kappa_{\text{start}} = \kappa_{\text{end}} = 0$ and $\beta_{\text{start}} = \beta_{\text{end}} = 1$, experimental setups used by Shi et al. (2021) and Mao et al. (2025) can be achieved, which only employ IBP or CROWN-IBP losses respectively. After the ramp-up phase, training is carried out over the full epsilon radius until it finishes. Regarding the number of epochs, we follow De Palma et al. (2024b) and train for 70 epochs on MNIST, 160 epochs for $\epsilon = \frac{2}{255}$ and 260 epochs for $\epsilon = \frac{8}{255}$ on CIFAR-10 and 160 epochs on TinyImageNet.

**Regularisation.** During the ramp-up phase, we employed the regulariser proposed by Shi et al. (2021) which is composed of two terms. One that penalises the explosion of IBP bounds during training time and one that balances inactive and active ReLU activations, *i.e.*, neurons that behave

only linearly and non-linearly for all inputs within the $\epsilon$ ball. The magnitude of this regularisation is controlled by two factors; a parameter $\lambda$ and a decay factor $1 - \frac{\epsilon}{\epsilon_{\text{train}}}$ with both of which the loss term is multiplied. This ensures that the regularisation is most prominently employed during the beginning of the training process which contributes to more stable training. In addition, we used $\ell_1$ regularisation weighed by a specified parameter. For its calculation, we exclusively considered the magnitude of weights in convolutional and linear layers in line with previous work (see, *e.g.*, Mao et al. (2025); De Palma et al. (2024b); Shi et al. (2021)).

**Optimisation.** We included the choice of an optimiser as well as the learning rate as part of our tuning scheme. Generally, we support *Adam* (Kingma & Ba, 2015), *AdamW* (Loshchilov & Hutter, 2019) as well as *RAdam* (Liu et al., 2020). We did not tune the internal hyperparameters of the optimisers, such as their $\beta$ values and weight decay, but used the defaults provided in PyTorch (Paszke et al., 2019). It is worth noting, that prior works did not consider different optimisers but exclusively relied on Adam for the optimisation; a choice not in line with advancements in the broader ML community (see, *e.g.*, Liu et al. (2022); Wightman et al. (2021)). For all conducted experiments, we employed a batch size of 512 while related work usually employed batch sizes of 256 on MNIST and 128 on CIFAR-10 and TinyImageNet (see, *e.g.*, Mao et al. (2025); De Palma et al. (2024b); Müller et al. (2023); Shi et al. (2021)). While we experienced in preliminary experiments that higher batch sizes do hurt the performance of certified training, we aimed to conduct our tuning using a higher batch size to fully exploit the capabilities of modern GPUs. In addition, our method searches for two epochs after ramp-up at which the learning rate is decayed by a given factor that is also optimised.

**Batch normalisation layers.** Shi et al. (2021) showed that BatchNorm layers are generally beneficial to the performance of certified training of deep neural networks. Therefore, we also employ them after every activation in the networks considered for our evaluation. In the literature, there are several options on how the statistics of the layers used to normalise batches should be set. Shi et al. (2021) and Müller et al. (2023) set the statistics based exclusively on unperturbed data, while De Palma et al. (2024b) use statistics over adversarial examples for the IBP bounds. At evaluation time, De Palma et al. (2024b) consider the statistics over both, perturbed and clean data. Mao et al. (2025) proposed to use statistics of unperturbed data for the PGD attack as well as for training. At test time, the authors employed statistics obtained over the whole population. Since multiple approaches exist and it is, to date, unclear whether any of them actually result in decisive performance differences, we chose to adopt the standard setting of CTRAIN that follows the approach of De Palma et al. (2024b) for SABR and MTL-IBP and the approach of Shi et al. (2021) for CROWN-IBP and IBP.

**Hyperparameter optimisation.** In our hyperparameter optimisation setup, we use the BoTorch (Balandat et al., 2020) sampler of Optuna (Akiba et al., 2019) with 10 initial random samples. We use a Gaussian Process as a surrogate model, with lengthscales as recommended by Hvarfner et al. (2024) and RBF kernel. The Gaussian Process hyperparameters are optimised using L-BFGS-B with marginal log likelihood loss. The inputs to the Guassian process are normalised to the range $[0, 1]$ and the target values are standardised. We optimise the acquisition function is optimised using L-BFGS-B. All design choices are based on the values found in (Akiba et al., 2019). Our hyperprameter optimisation method does not use any previously known configurations or priors, making the optimisation procedure generalisable for new, unseen scenarios.

### B.5 ADDITIONAL IMPLEMENTATION DETAILS

To run our optimisation method, we relied on CTRAIN (Kaulen & Hoos, 2025) in version `0.4.2` for the implementation of the certified training methods. CTRAIN includes implementations of several state-of-the-art methods, including the methods investigated in this work, as well as the proposed initialisation and regularisation procedures of Shi et al. (2021). Further, it implements IBP, CROWN-IBP and CROWN (Zhang et al., 2018) for incomplete verification and the adversarial attack PGD (Madry et al., 2018) for fast disproving of robustness. For the bounding process and incomplete verification, CTRAIN in turn relies on the `auto_LiRPA` library (Xu et al., 2020) at commit `cf0169c`. Lastly, the neural network training is carried out using PyTorch (Paszke et al., 2019) in version `2.3.1`.

### B.6 Complete Verification

For complete verification, we used the state-of-the-art (Brix et al., 2024; König et al., 2024) complete verification system $\alpha\beta$-CROWN (Wang et al., 2021; Xu et al., 2021; Zhang et al., 2018). While it is known, that careful parameter tuning of $\alpha\beta$-CROWN is crucial to obtain strong results, we used the system in its standard configuration to not create a biased evaluation, where one certified training method or network architecture might benefit more from the selected parameter choices. We set the batch size of Branch-and-Bound domains to the highest number our hardware could accomodate, resulting in a batch size of 1024 for CNN7, Narrow CNN7 and CNN5 and a batch size of 512 for CNN7 Wide and CNN9 on CIFAR-10. We used a batch size of 1024 for MNIST and 16 for verifying networks trained on TinyImageNet. We used a cutoff time of 1 000s in wall-clock time for verification of CNN7 on CIFAR-10 and TinyImageNet. For MNIST and the results on additional architectures presented later, we used a cutoff of 300s in wall-clock time to keep computational demands manageable.

### B.7 Configuration Spaces

The configuration spaces used in our experiments are shown in Table 3. Each space consists of a set of base hyperparameters shared across all methods, extended with method-specific ones where necessary. In the following, we provide a brief explanation of each hyperparameter included. Generally, we ensured in our design of the search space that it encompasses all previously chosen parameter values from the literature but also includes all sensible parameter choices to allow for the discovery of novel, well-performing configurations.

**Warm up epochs** refer to the number of epochs for which the network is trained on clean cross entropy loss at the beginning of the training schedule.

**Ramp-up epochs** refer to the previously explained training phase, where $\epsilon$ is annealed from 0 to its final value. We employ 10 such epochs at least and make the maximum number dependent on the number of total epochs the network should be trained for, thereby making the search space flexible and applicable to new benchmarks. At most, we extend the ramp-up phase through 75% of the total number of epochs. This way, the ramp-up phase will have completed at the end of training, even when the maximally allowed warm- and ramp-up durations are chosen.

**LR decay factor** describes the factor by which the learning rate is decayed at up to two epochs after the ramp-up phase, for which we also optimise.

**LR decay epoch {1,2}** describe the points in time at which the learning rate is decayed. We calculate the first point by adding *LR decay epoch 1* to the number of warm- and ramp-up epochs, ensuring that the learning rate is only decayed after the ramp-up phase completed. The second point is calculated analogously, by adding the value of *LR decay epoch 2* to the epoch at which the learning rate was decayed first. If any of these decay epochs exceed the total number of training epochs, they are ignored.

**L1 regularisation weight** refers to the weight with which L1 regularisation is employed during training.

**Shi regularisation weight** refers to the $\lambda$ parameter which refers to the magnitude of the regularisation proposed by Shi et al. (2021) during the ramp-up phase.

**Train $\epsilon$ factor** scales the $\epsilon$ value the network is evaluated on by a given factor for training. In some cases, this has shown to be beneficial (see, *e.g.*, Gowal et al. (2019); Shi et al. (2021).

**Optimiser** refers to the choice of the optimiser used for the training procedure. We include *Adam* (Kingma & Ba, 2015), *AdamW* (Loshchilov & Hutter, 2019) and *RAdam* (Liu et al., 2020).

**Learning rate** refers to the initial learning rate employed by the previously chosen optimiser.

**Start & end $\kappa$** refer to the $\kappa$ value employed in IBP and CROWN-IBP to weigh standard cross-entropy loss and the certified loss. During the ramp-up phase, $\kappa_{\text{start}}$ is gradually decreased to $\kappa_{\text{end}}$, placing greater weight on the natural loss in the early stages to stabilise training before progressively shifting the focus toward the certifiability objective. To ensure that $\kappa_{\text{start}}$ always exceeds $\kappa_{\text{end}}$, we

Table 3: Configuration spaces employed in our hyperparameter optimisation method for certified training. Square brackets indicate continuous parameters for which we give inclusive upper and lower limits. Curly brackets indicate sets out of which the optimiser can choose one option. Finally, single numbers indicate constants, *i.e.*, parameters that remain unchanged throughout the hyperparameter optimisation.

| Method | Hyperparameter | Range |
|---|---|---|
| All | Warm up epochs | $[0, 5]$ |
| | Ramp up epochs | $[10, 0.75 \cdot \text{Total Epochs}]$ |
| | LR decay factor | $[\text{1e-5}, 0.9]$ |
| | LR decay epoch 1 | $[0, 0.5 \cdot \text{Total Epochs}]$ |
| | LR decay epoch 2 | $[0, 0.25 \cdot \text{Total Epochs}]$ |
| | L1 regularisation weight | $[\text{1e-8}, \text{1e-4}]$ |
| | Shi regularisation weight | $[0.0, 1.0]$ |
| | Train $\epsilon$ factor | $[1.0, 2.0]$ |
| | Optimiser | $\{\text{Adam, AdamW, RAdam}\}$ |
| | Learning rate | $[\text{1e-5}, \text{1e-1}]$ |
| IBP | Start $\kappa$ | $[0, 1]$ |
| | End $\kappa$ | $[0, 1]$ |
| CROWN-IBP | Start $\kappa$ | $[0, 1]$ |
| | End $\kappa$ | $[0, 1]$ |
| | Start $\beta$ | $1.0$ |
| | Start $\beta$ | $[0, 1]$ |
| SABR | $\tau$ | $[0.001, 0.5]$ |
| | PGD steps | $[1, 10]$ |
| | PGD step size | $[0.1, 2]$ |
| | PGD restarts | $1$ |
| | PGD $\epsilon$ scaling factor | $[1, 3]$ |
| MTL-IBP | $\alpha$ | $[0.001, 0.5]$ |
| | PGD steps | $[1, 10]$ |
| | PGD step size | $[0.1, 2]$ |
| | PGD restarts | $1$ |
| | PGD $\epsilon$ scaling factor | $[1, 3]$ |

define the latter as a multiplicative factor $c$ of the former, *i.e.*, $\kappa_{\text{end}} = \kappa_{\text{start}} \times c$ and optimise the factor $c$ instead of optimising $\kappa_{\text{end}}$ directly.

**Start & end** $\beta$ are handled analogously, but we fix $\beta_{\text{start}} = 1.0$ to ensure that the full benefit of the tighter relaxation used in CROWN-IBP is employed to stabilise early training stages.

$\tau$ refers to the subselection ratio used in SABR that weighs certified with adversarial loss (De Palma et al., 2024b; Müller et al., 2023).

$\alpha$ refers to the parameter of MTL-IBP that weighs certified with adversarial loss (De Palma et al., 2024b).

**PGD steps, step size, restarts and $\epsilon$ scaling factor** refer to the parameters of the adversarial attack employed during training to approximate the adversarial loss (Madry et al., 2018). Here, steps specify the number of optimisation steps, while step size indicates the magnitude of the input change allowed per iteration. To keep training costs tractable, we chose to always randomly initialise the attack once within the $\epsilon$ ball and not multiple times as done by Mao et al. (2025); a choice consistent with multiple other works in the field (De Palma et al., 2024b; Müller et al., 2023; Madry et al., 2018). This strategy leverages the fact that each training sample is reinitialised differently across epochs, yielding a good approximation of the worst-case adversarial loss overall. Finally, we optimise a factor that scales the $\epsilon$ radius in the adversarial attack, increasing the emphasis on the adversarial loss when combined with the certified loss (De Palma et al., 2024b), achieving a similar effect to ReLU shrinking as used by Müller et al. (2023).

Table 4: Comparison of the results reported from the literature to the results achieved by using our novel optimisation procedure on MNIST with $\epsilon = 0.3$. For each result from the literature, we selected a configuration from the Pareto front that achieves similar or better performance. Boldface marks results surpassing prior work; underlined values indicate similar performance ($\pm 0.5$).

| Dataset | $\epsilon$ | Method | Source | Clean Acc. [%] (Lit.) | Cert. Acc. [%] (Lit.) | Clean Acc. [%] (ours) | Cert. Acc. [%] (ours) |
|---|---|---|---|---|---|---|---|
| MNIST | 0.3 | MTL-IBP | De Palma et al. (2024b) | 98.80 | 93.62 | 98.66 | 93.73 |
| | | | Mao et al. (2025) | 98.74 | 93.90 | 98.66 | 93.73 |
| | | SABR | Müller et al. (2023) | 98.75 | 92.98 | 98.77 | 93.43 |
| | | | Mao et al. (2025) | 98.66 | 93.68 | 98.75 | 93.55 |
| | | IBP | Shi et al. (2021) | 97.67 | 93.10 | **98.55** | **93.89** |
| | | | Mao et al. (2025) | 98.54 | 93.80 | 98.52 | 94.00 |
| | | CROWN-IBP | Xu et al. (2021) | 98.18 | 92.98 | 97.98 | 93.22 |
| | | | Mao et al. (2025) | **98.48** | **93.90** | 97.94 | 93.25 |

## C  ADDITIONAL EXPERIMENTS

In the following, we give results of experiments conducted on additional datasets and architectures.

### C.1  ADDITIONAL DATASETS

We evaluated our approach on MNIST (LeCun, 1998) with $\epsilon = 0.3$, following the experimental setup outlined earlier. While we left our optimisation procedure unchanged, we ran verification with a cutoff time of 300s to reduce the computational burden. Nevertheless, we believe that our results regarding certified accuracy could be further strengthened when employing cutoff times of 1 000 seconds as done in related work (see, *e.g.*, (Mao et al., 2025; De Palma et al., 2024b)). We show the Pareto fronts found using our novel method in Figure 7 and compare to results from the literature in Table 4. While our method generally achieves comparable performance to configurations reported in the literature, it did not identify configurations that substantially surpass prior results. We attribute this to the fact that current certified training techniques have likely already been tuned to the maximal performance achievable with IBP-based training for the given benchmark. This hypothesis is reinforced by the observation that all methods converge to very similar trade-offs in our analysis, suggesting that a performance barrier has likely been reached. However, the fact that we were able to retrieve these high-performing configurations underlines the effectiveness of our method once more.

### C.2  ADDITIONAL ARCHITECTURES

Recently, Mao et al. (2024) showed, both theoretically and empirically, that architecture, specifically network depth and width, has a major impact on the performance of certified training techniques.

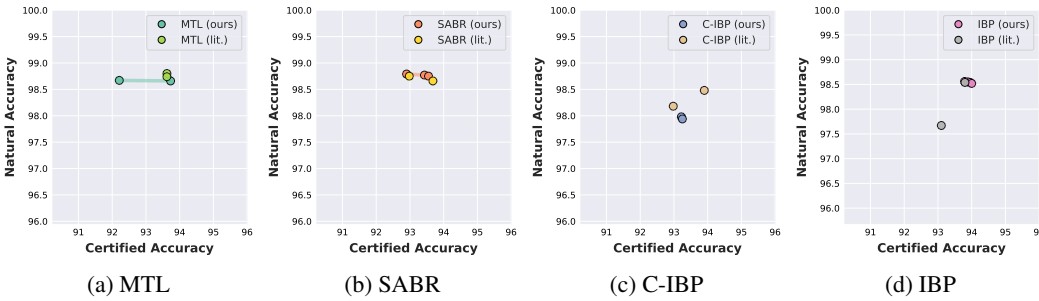

(a) MTL  (b) SABR  (c) C-IBP  (d) IBP

Figure 7: Results for MNIST with $\epsilon = 0.3$ yielded by our method. We compare Pareto fronts obtained using our method to results given in the original publications and the recent CTBench benchmark (Mao et al., 2025).

Table 5: Comparison of performances on CIFAR-10, $\epsilon = \frac{2}{255}$, of well-performing configurations reported by the respective authors across different codebases. We retrain all configurations using CTRAIN (Kaulen & Hoos, 2025) and compare them to the results reported in the original publications. CTRAIN achieves similar results across all methods, revealing that advancements achieved by our method cannot be traced back to the employed implementation.

| Dataset | $\epsilon$ | Method | Source | Clean Acc. [%] (Lit.) | Cert. Acc. [%] (Lit.) | Clean Acc. [%] (CTRAIN) | Cert. Acc. [%] (CTRAIN) | Adv. Acc. [%] (CTRAIN) |
|---|---|---|---|---|---|---|---|---|
| CIFAR-10 | $\frac{2}{255}$ | MTL-IBP | De Palma et al. (2024b) | 80.11 | 51.35 | 80.04 | 50.09 | 68.76 |
| | | SABR | Müller et al. (2023) [*] | 79.24 | 62.84 | 79.66 | 46.29 | 64.06 |
| | | IBP | Shi et al. (2021) | 66.84 | 52.85 | 67.35 | 53.21 | 57.50 |
| | | CROWN-IBP | Zhang et al. (2020) [†] | 71.52 | 53.97 | 67.26 | 53.97 | 57.82 |
| | $\frac{8}{255}$ | MTL-IBP | De Palma et al. (2024b) | 53.35 | 34.64 | 54.34 | 32.33 | 38.06 |
| | | SABR | Müller et al. (2023) [*] | 52.38 | 35.13 | 51.67 | 34.47 | 38.77 |
| | | IBP | Shi et al. (2021) | 48.94 | 34.97 | 48.04 | 33.63 | 36.93 |
| | | CROWN-IBP | Zhang et al. (2020) [†] | 46.29 | 33.38 | 46.83 | 33.13 | 35.68 |

[*]: Results were obtained with complete verification.
[†]: Results were obtained without improvements by Shi et al. (2021) and a longer training schedule.

The authors found that the CNN7 Wide network defined earlier exhibits optimal depth and width for certified training techniques. We investigated whether this claim still holds when considering a Pareto front as the performance measure by running our novel method on CNN5, CNN7 Wide, CNN7 Narrow and CNN9 using the CIFAR-10 dataset with $\epsilon = \frac{2}{255}$. However, since running complete verification for all networks would incur substantial computational costs, we opted for a preliminary experiment where we only verified the first $1\,000$ images of the test set with a cutoff time of 300s. We present the resulting Pareto fronts in Figure 8. Our analysis reveals that, indeed, the CNN7 Wide architecture yields very strong trade-offs across the performance space. However, there are also other architectures that contribute to a combined Pareto front over all architectures. For MTL-IBP, the Pareto front also includes two CNN5 models, whereas for CROWN-IBP it includes one CNN5 model. For SABR, 50% of the Pareto front consists of the standard CNN7 architecture, particularly for configurations targeting higher certified accuracies. Finally, for standard IBP training, a single CNN9 model appears on the Pareto front, achieving a trade-off comparable to that of the CNN7 Wide models. This preliminary experiment highlights that our Pareto front analysis may reveal previously unknown performance complementarities regarding different architectures and motivates future work.

## C.3 COMPARISON TO CONFIGURATIONS REPORTED IN THE LITERATURE

To ensure that our reported performance gains are not due to the different codebase used for the experiments, we train with configurations reported in the literature as best-performing using CTRAIN (Kaulen & Hoos, 2025). For this, we consider configurations for SABR and MTL-IBP from their original publications and configurations for IBP and CROWN-IBP from Shi et al. (2021). We trained those on CIFAR-10 with $\epsilon = \frac{2}{255}$ and evaluated them using incomplete verification, *i.e.*, CROWN (Zhang et al., 2018). We also provide adversarial accuracy as an upper bound to the

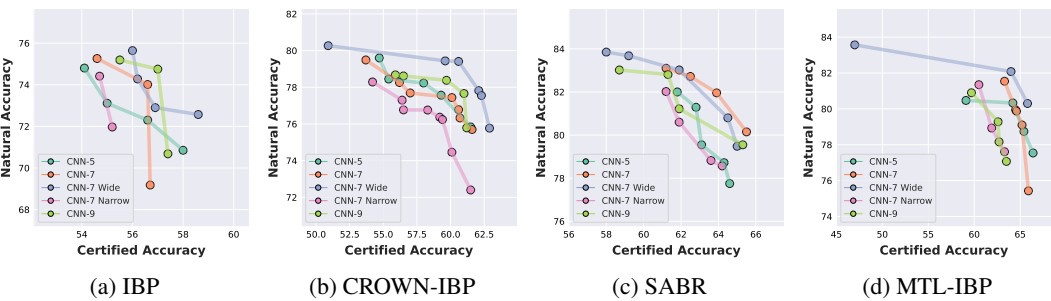

    (a) IBP          (b) CROWN-IBP         (c) SABR          (d) MTL-IBP

Figure 8: Pareto fronts on CIFAR-10 with $\epsilon = \frac{2}{255}$ yielded by our method for the architectures CNN5, CNN7, CNN7 Wide, CNN7 Narrow as well as CNN9.

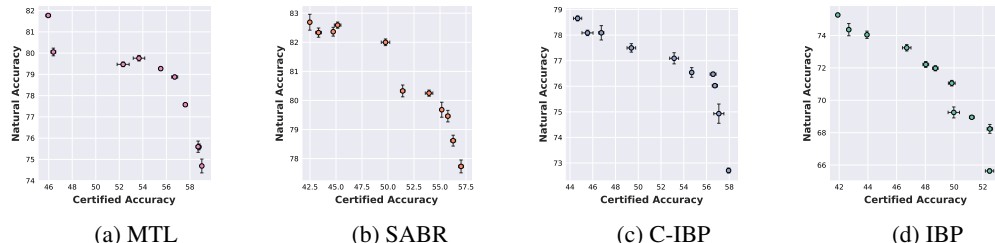

| (a) MTL | (b) SABR | (c) C-IBP | (d) IBP |

Figure 9: Pareto fronts obtained using our method on CIFAR-10 with $\epsilon = \frac{2}{255}$ with error bars. Each dot represents the average performance over three pseudo-random seeds and error bars indicate standard deviation.

certified robustness achievable through complete verification. In Table 5 we compare the obtained results to those reported in the literature. It is important to note, that the authors of SABR did not provide results on incomplete verification (Müller et al., 2023) and Shi et al. (2021) did not provide results on CROWN-IBP. Thus, we compare to results obtained using complete verification on SABR and to results without the improvements by Shi et al. (2021) and a longer training schedule on CROWN-IBP. The experiment shows that CTRAIN achieves similar results to the ones reported by the original authors with negligible differences. Therefore, we conclude that the success of our method cannot be attributed to the used codebase and might most probably work well when using other implementations as well.

### C.4 VARIANCE OF RESULTS

In this experiment we evaluate each configuration resulting from the hyperparameter optimisation procedure using three pseudo-random seeds to assess result variance. We present the outcomes in Figure 9, where each data point represents the mean and error bars indicate standard deviation.

It is important to note, that the algorithm performances resulting from this experiment do not create a Pareto front, since some of the configurations dominate others. The reason for this is the fact that we evaluate each configuration only once during the HPO procedure, which is a known practice when optimising neural network hyperparameters due to the high training cost associated with it (see, *e.g.*, Zela et al. (2022)). This setup allows "lucky" configurations to appear on the Pareto front, while "unlucky" ones may be excluded even if their average performance would place them on the front. Therefore, the hyperparameter optimisation might overfit to the chosen training seed. This issue is further compounded by the inherent non-determinism of GPU-based neural network training, which can lead to noticeable performance differences even with the same training seed. A strongly related topic to this issue is overtuning in hyperparameter optimisation, an active area of research (see, *e.g.*, Schneider et al. (2025); Nagler et al. (2024)). One method to mitigate these phenomena is evaluating each configuration multiple times during optimisation and using its average performance, which is computationally infeasible given the costs of certified training.

However, we strongly believe that this does not undermine our results. Our final evaluation is performed on an unseen test set, and we consider differences significant only if they exceed $\pm 0.5$ compared to previously known results. This threshold corresponds to the maximum standard deviation observed when training each method with three seeds.

### D COMPUTATIONAL COSTS

We present the computational costs for both the hyperparameter optimisation runs and the verification in Table 6. We display the required total compute for discovery and complete verification of the Pareto front as well as the average verification time per instance. We show that hyperparameter optimisation costs scale directly with training costs, with TinyImageNet being the most expensive benchmark due to its larger scale. Across the same dataset, architectures with fewer parameters incur lower optimisation costs, with CNN5 being the cheapest to optimise. However, regarding the costs of complete verification, perhaps surprisingly, the highest costs occur on the CIFAR-10 dataset with $\epsilon = \frac{2}{255}$. We attribute this to the fact that, on this benchmark, complete verification

Table 6: Computation time of our experiments in wall-clock time. For each experiment, we show the time required for the hyperparameter optimisation, the average verification time required per instance as well as the total time used for complete verification of the Pareto front. If not indicated otherwise, we report verification times over the complete test-set with a per-instance timeout of 1 000s in wall-clock time.

| Dataset | Network | Method | $\epsilon$ | HPO (h) | Verification (s) Average | Verification (h) Total |
|---------|---------|--------|-----------|---------|---------|---------|
| CIFAR10 $^{\star\dagger}$ | CNN5 | MTL-IBP | $\frac{2}{255}$ | 113.69 | 10.96 | 33.48 |
| | | SABR | | 111.02 | 9.18 | 28.06 |
| | | CROWN-IBP | | 135.15 | 6.09 | 15.23 |
| | | IBP | | 62.95 | 2.23 | 6.21 |
| CIFAR10 | CNN7 | MTL-IBP | $\frac{2}{255}$ | 236.68 | 47.34 | 1314.97 |
| | | SABR | | 226.83 | 52.03 | 1387.50 |
| | | CROWN-IBP | | 318.13 | 24.66 | 704.08 |
| | | IBP | | 95.70 | 7.39 | 225.95 |
| CIFAR10 | CNN7 | MTL-IBP | $\frac{8}{255}$ | 340.25 | 10.07 | 378.07 |
| | | SABR | | 296.88 | 15.54 | 461.29 |
| | | CROWN-IBP | | 457.43 | 9.10 | 202.11 |
| | | IBP | | 158.81 | 13.69 | 436.11 |
| CIFAR10 $^{\star\dagger}$ | CNN9 | MTL-IBP | $\frac{2}{255}$ | 339.96 | 17.64 | 53.89 |
| | | SABR | | 336.08 | 18.80 | 67.89 |
| | | CROWN-IBP | | 434.99 | 8.15 | 15.85 |
| | | IBP | | 140.74 | 3.77 | 15.71 |
| CIFAR10 $^{\star\dagger}$ | Narrow CNN7 | MTL-IBP | $\frac{2}{255}$ | 182.16 | 15.99 | 44.41 |
| | | SABR | | 172.72 | 13.44 | 44.80 |
| | | CROWN-IBP | | 211.36 | 5.27 | 19.03 |
| | | IBP | | 75.45 | 2.63 | 8.03 |
| CIFAR10 $^{\star\dagger}$ | Wide CNN7 | MTL-IBP | $\frac{2}{255}$ | 409.45 | 27.56 | 53.59 |
| | | SABR | | 399.83 | 21.98 | 79.38 |
| | | CROWN-IBP | | 624.31 | 12.66 | 31.64 |
| | | IBP | | 164.31 | 4.62 | 17.98 |
| MNIST $^{\star}$ | CNN7 | MTL-IBP | 0.3 | 117.15 | 5.02 | 27.91 |
| | | SABR | | 104.22 | 4.79 | 93.22 |
| | | CROWN-IBP | | 140.0 | 2.05 | 22.79 |
| | | IBP | | 51.22 | 3.16 | 43.91 |
| TinyImagenet | CNN7 | MTL-IBP | $\frac{1}{255}$ | 1576.51 | 37.43 | 207.96 |
| | | SABR | | 1494.89 | 45.68 | 888.30 |
| | | CROWN-IBP | | 1567.99 | 15.12 | 209.96 |
| | | IBP | | 757.65 | 16.32 | 408.07 |

$^{\star}$: Selected networks were verified with a per-instance timeout of 300 seconds in wall-clock time.
$^{\dagger}$: For selected networks, we report verification times over the first 1000 images of the test set.

methods achieve the largest improvements compared to cheaper, incomplete methods. On the other benchmarks, incomplete methods are often sufficient to certify most provably robust instances.

# E  ADDITIONAL DISCUSSION

In the following, we discuss several of our design decisions in developing our novel method and give rationale on the selection of methods included in the evaluation.

## E.1  MOTIVATION FOR MULTI-OBJECTIVE OPTIMISATION

In related work, the hyperparameter optimisation problem has been treated as a single-objective problem with optimising for certified accuracy only (Mao et al., 2025) or by optimising for the

sum of clean and certified accuracy (De Palma et al., 2024b). This circumstance already highlights that there exist multiple views on what defines a well-performing configuration with regard to the robustness-accurcay trade-off (*e.g.*, given two configurations with performances $(0.8, 0.63)$ and $(0.78, 0.64)$, De Palma et al. (2024b) would prefer the former while Mao et al. (2025) would choose the latter). Therefore, there does not exist a clear single-objective definition of strong performance for certified training. Thus, we opted to provide a method that approximates the whole Pareto set of configurations with strong robustness-accuracy trade-offs. It is important to note that, given our method, a potential user can constrain the optimisation to regions that are important to them, *e.g.* prioritising strong certifiability over clean performance.

In addition, we argue that the multi-objective approach enabled successful automated hyperparameter optimisation for certified training of deep neural networks in the first place. As mentioned previously, conducting complete verification for all investigated configurations during optimisation is computationally infeasible. Thus, we optimised for a proxy metric, *i.e.*, certified accuracy obtained through cheaper incomplete verification methods. However, we found that the Pareto front often included configurations with similar sums of performances, *e.g.*, $(0.8, 0.5)$ vs. $(0.75, 0.55)$, where higher certifiability with incomplete methods only led to faster complete verification but not to higher certified accuracies overall. In these cases, the former configuration should be preferred since it yields a generally better trade-off. However, if the optimisation objective were the sum of certified and clean accuracy, the optimiser could not distinguish between the two; if only certified accuracy were used, it would favour the latter. With our method, both configurations are included in the Pareto set and considered for complete verification, ultimately revealing the superior performance of the former configuration. We therefore conclude that a multi-objective approach is essential for efficiently identifying the best-performing configurations.

### E.2 JOINT HYPERPARAMETER-OPTIMISATION

While prior work (Mao et al., 2025; De Palma et al., 2024b; Müller et al., 2023) showed that the robustness-accuracy trade-off can be explored by tuning method-inherent trade-off parameters such as $\alpha$ for MTL-IBP and $\tau$ for SABR, we decided to investigate the trade-off through joint optimisation of all relevant parameters of the training pipeline, including method-specific as well as general deep learning parameters. First and foremost, it is a well-known fact that hyperparameters often exhibit complex interactions and that it is therefore required to optimise all parameters jointly to identify best-performing configurations (Hutter et al., 2014b). In the context of certified training, our analysis in Appendix A revealed that several hyperparameters contribute strongly to overall performance. For example, the choice of the optimiser used in SABR or the number of warm-up epochs employed in MTL-IBP were crucial to obtain strong trade-offs on CIFAR-10 with $\epsilon = \frac{2}{255}$. These well-performing configurations could not have been discovered when tuning only single parameters.

### E.3 RATIONALE FOR EXCLUSION OF TAPS AND STAPS

While we included several state-of-the-art methods in our evaluation, we decided against considering the recently proposed TAPS and STAPS certified training methods (Mao et al., 2023). These methods train by propagating interval (TAPS) or SABR (STAPS) bounds through a predefined number of layers, and then performing adversarial training in the latent space within the resulting bounds for the remaining layers of the network. Mao et al. (2023) demonstrated that these methods can achieve strong performance on standard benchmarks of the certified training community. However, both the original evaluation of the authors (Mao et al., 2023) as well as the recent CTBench benchmark (Mao et al., 2025) showed that neither TAPS nor STAPS outperforms MTL-IBP.

While we would have preferred to include these methods in our evaluation to assess their performances using Pareto front analysis, we faced several challenges. First, the choice of network split has a major impact on method performance, with almost all splits except the best-performing one yielding sub-par or even catastrophic results. We therefore assume that proper hyperparameter optimisation would require far more trials than used in our experiments. Additionally, TAPS and STAPS achieve their best performance when paired with strong latent-space adversarial attacks with multiple random restarts (Mao et al., 2025; 2023). STAPS also requires an input-space adversarial attack to compute SABR bounds. Consequently, TAPS and STAPS are generally very costly, making efficient hyperparameter optimisation even more challenging. In conclusion, we chose to

exclude TAPS and STAPS from this study but plan to investigate their performance with regard to the obtained Pareto fronts in future work.

# F PSEUDO-CODE

Algorithm 1 provides pseudo-code for our proposed constrained multi-objective hyperparameter optimisation method for certified training of deep neural networks. Line 1 gathers the initial random samples, which are evaluated in line 2. In line 4, we determine which configurations belong to the Pareto set. The optimisation loop then begins with fitting the surrogate models in line 6. Line 7 then optimises the acquisition function to decide on the next candidate configuration. We then evaluate the configuration and add it to the set of evaluated configurations in line 8. Lastly, in line 9, we determine which configurations belong to the Pareto set based on the updated set of evaluated configurations.

---

**Algorithm 1** Multi-objective hyperparameter optimisation for certified training

---

1: **Input:** total budget $b$, initial sample size $r$, certified training method $t$, incomplete verification method $v$, dataset $D$, min. clean acc. constraint $c_{\text{clean}}$, min. cert acc. constraint $c_{\text{cert}}$
2: Initialise $\zeta$ with $r$ randomly sampled points
3: $\zeta \leftarrow \{(\lambda, v(t(D, \lambda))) \mid \lambda \in \zeta\}$
4: $P \leftarrow \{(\lambda, (m_{\text{clean}}, m_{\text{cert}})) | \nexists_{(\lambda', (m'_{\text{clean}}, m'_{\text{cert}})) \in \zeta}(m_{\text{clean}}, m_{\text{cert}}) \prec (m'_{\text{clean}}, m'_{\text{cert}}) \wedge (m_{\text{clean}} \geq c_{\text{clean}} \wedge m_{\text{clean}} \geq c_{\text{clean}})\}$
5: **while** budget $b$ is not exhausted **do**
6:     $S_{\text{clean}}, S_{\text{cert}}, S_{\text{clean cond}}, S_{\text{cert cond}} \leftarrow \text{fit}(\zeta)$
7:     $\lambda_t \leftarrow \arg\max \text{EHVI}(S_{\text{clean}}, S_{\text{cert}}, S_{\text{clean cond}}, S_{\text{cert cond}}, P, c_{\text{clean}}, c_{\text{cert}})$
8:     $\zeta \leftarrow \zeta \cup \{(\lambda, v(t(D, \lambda)))\}$
9:     $P \leftarrow \{(\lambda, (m_{\text{clean}}, m_{\text{cert}})) | \nexists_{(\lambda', (m'_{\text{clean}}, m'_{\text{cert}})) \in \zeta}(m_{\text{clean}}, m_{\text{cert}}) \prec (m'_{\text{clean}}, m'_{\text{cert}}) \wedge (m_{\text{clean}} \geq c_{\text{clean}} \wedge m_{\text{cert}} \geq c_{\text{cert}})\}$
10: **end while**
11: **Return** $P$

---

