# OpenReview forum: "Empirically Investigating the Trade-Offs in Deterministic Certified Training"
_ICLR.cc/2026/Conference — Submitted to ICLR 2026_

### Official Review · Reviewer_ECKU · 2025-10-15

**Soundness:** 2
**Presentation:** 3
**Contribution:** 2
**Rating:** 4
**Confidence:** 5

**Summary:**

This paper studies the hyperparameter-tuning problem in certified training, which is of significance as certified training algorithms typically have much more hyperparameters than other training paradigms.

On the positive side, this paper proposes to use Bayesian optimization on the hyperparameter space instead of grid search as applied by previous works, which has been proven effective both by prior works in other domains and this work in the certified training domain. The method yields a Parento frontier rather than a single model tuned exclusively for maximizing the certified accuracy metric, thus is of potential interest to the community.

On the negative side, even with a much larger hyperparameter space, e.g., choice of optimizers, lr, decay epoch, random seeds, etc., the improvement over the grid search is marginal on SOTA. For example, on cifar 2/255, the certified accuracy was 64.54% compared to prior work which is 64.41%. The large improvement on TinyImagenet is particularly encouraging though, as 30.67% compared to 27.73%, which is of particular interest.

Overall, this paper brings the insights from autoML to certified training, including existing techniques and demonstrating similar advantages, which is interesting but might be of limited novelty.

**Strengths:**

This paper is clearly written and the idea of using Bayesian optimization to conduct hyperparameter search in a larger space more effectively makes sense.

The improvement on TinyImagenet is interesting, especially comparing against the minimal improvement in other settings.

**Weaknesses:**

The main limitation might be the novelty. AutoML, especially methods to autonomously tune hyperparameters with methods such as Bayesian optimization, is a textbook algorithm. There is no specific challenge in applying these methods to certified training, as have done by this work. As far as I am concerned, the only reason grounding this work could be an empirical illustration and evaluationg of a specific auto-tuning algorithm. If considered this way, then the contribution of this work should include more insights in applying auto-tuning, especially those novel in certified training compared to other domains.

Another major but fixable limitation is the experimental comparison. In Table 1, all comparisons with Xu et al and Zhang et al on CROWN-IBP should be removed. This is because these prior works use a very different model architecture, while the results in other works and this work uses the SOTA CNN7 architecture, which is identified independently. Further, the comparison to Mao et al on CROWN-IBP in CIFAR-10 is unfair: they use loss fusion universally, while this work did not apply loss fusion. As argued by Mao et al, this is due to practical concerns since CROWN-IBP without loss fusion does not scale to large number of classes, but not resulted from certain limitation of grid search tuning. This is supported by the vanishing improvement on CROWN-IBP in TinyImagenet, where this work applied loss fusion as well. Formatting Table 1 in its current form misleads readers to believe the relatively large gain in CIFAR comes from auto-tuning, while it is not. In addition, Table 1 should only compare against the result in Mao et al, but not the original works; as discussed by them, the original works under-tune the hyperparameter and have problematic implementations, thus comparing against those does not yield meaningful conclusions, especially with regard to the benefit of auto-tuning against grid search. When compares against Mao et al., it is worth noting that this work further tunes on random seeds, while they fix the random seed throughout CTBench (Appendix A). A comparison involving only a single random seed should be placed in the appendix to provide a fair comparison between auto-tuning and grid-search.

Some related work in certified training is missing: In Sec 3 part 1, [1,2,3] should be placed in the context of SOTA algorithms as well. Since certified training is of central study, recent works in certified training should be discussed: [4] prove that certified training does not allow single-neuron convex relaxations to yield exact bounds; [5] prove that certified training allows multi-neuron convex relaxations to exactly bound every piecewise linear continuous function; [6] empirically studies certified training in empirical robustness. This might not be a complete list.

In addition, using Bayesian optimization to auto-tune hyperparameters faces a major obstacle in parallism. While grid-search can be effectively executed in parallel and thus does not incur additional wall-clock time overhead, Bayesian optimization, despite possibly reducing the total number of trials, is bounded to execute sequentially and thus leads to larger waiting time.

Finally, the authors should remark clearly on the implications for the field. The community is interested in pursuing new frontiers, as witnessed by new SOTA. Independently, if a good set of hyperparameters can be identified, then it saves efforts in future works in hyperparameter tuning. Therefore, the author should clearly present their insights in the resulting hyperparameters, especially on how they could help the community without running 300 trials per benchmark as in this work. Further, it is **very meaningful** to discuss why SOTA (MTL-IBP) in TinyImagenet witnessed particularly large gains.

Minor comments: The authors discussed grid-search as "labour-intensive", while it is only computationally expensive. A grid search hardly requires expertise as well. The argument in excluding TAPS is problematic: as shown by CTBench, TAPS running time is usually comparable, up to 2x cost with regard to other methods, and it exceeds MTL-IBP on MNIST although it is not highlighted by this work in the main text. In addition, some hyperparameters are discrete, e.g., the choice of optimizers; the authors should clearly discuss how these discrete variables are handled by the Gaussian process.

[1] https://arxiv.org/abs/2305.04574

[2] https://arxiv.org/abs/2206.14772

[3] https://arxiv.org/abs/2403.07095

[4] https://arxiv.org/abs/2311.04015

[5] https://arxiv.org/abs/2410.06816

[6] https://arxiv.org/abs/2410.01617

**Questions:**

Please address questions raised in weaknesses.

Further, why does Table 1 has different numbers for the comparison against Mao et al.? What is changed in the numbers presented in this work? My guess is that the authors picked different points on the Parento frontier, is it correct?

---

> ### Author Response · Authors · 2025-11-20
> **Rebuttal [1]**
>
> We thank the reviewer for their thoughtful and detailed review. We are particularly encouraged that the reviewer recognised the significance of automated hyperparameter optimisation for certified training since “those training algorithms typically have much more hyperparameters than other training paradigms”. Furthermore, we are delighted that the conducted analysis of Pareto fronts of configurations as well as the application of sophisticated hyperparameter optimisation techniques to certified training was identified as potentially interesting to the community. In addition, we are happy to hear that the improvements yielded by our method on TinyImageNet were recognised as particularly compelling. In the following, we address the potential weaknesses of our paper mentioned by the reviewer:
>
> **Q: The submission exhibits limited novelty since there is no specific challenge in applying Bayesian optimisation to certified training.**
>
> We regret that the paper might not have sufficiently highlighted the challenges encountered and solved by us in applying Bayesian optimisation to certified training. We will address them in a revised version of the manuscript. Automated hyperparameter optimization methods cannot be directly applied to IBP-based certified training due to conflicting objectives between certifiability and standard performance, the computational intractability of complete verification for each configuration, and the absence of expert-designed search spaces. We address these challenges and make the first successful application of automated hyperparameter optimisation in this context. More specifically, we employ multi-objective optimisation, which allows the optimizer to distinguish qualitative differences between configurations that achieve the same sum of objectives. We show that certifiability measured using incomplete verification serves as an efficient, low-cost proxy for identifying best-performing configurations. Additionally, we provide a novel search space that jointly incorporates both general deep learning hyperparameters and method-specific parameters. In doing this, we further introduce a novel evaluation method for certified training by comparing Pareto fronts, providing a nuanced assessment of the trade-offs achievable across the entire objective space.
>
> **Q: Improvement over the state of the art is marginal on CIFAR-10.**
>
> We respectfully disagree. While it is true that the best achievable certified robustness on CIFAR-10, $\epsilon=\frac{2}{255}$ using our method does not significantly exceed the highest certified robustness reported in the literature, our method identifies a configuration that achieves similar certified robustness while providing a substantial improvement of approximately 1% in clean accuracy. Furthermore, we demonstrated that SABR can achieve over 2% higher clean accuracy while maintaining certifiable accuracies comparable to MTL-IBP, which we consider a highly significant result for the community. In general, we demonstrated across nearly all scenarios that similar or better certifiable guarantees can be achieved while improving performance on clean data. This is a highly relevant finding for the real-world application of IBP-based models, where both clean-data performance and certifiable guarantees are important.
>
> **Q: Table 1 should not compare to the original CROWN-IBP publications.**
>
> We acknowledge that Zhang et al. [1] and Xu et al. [2] use a different architecture than the one employed in our study. However, our goal was to compare our results to the best results known in the literature with regard to natural and certified accuracy. We will update the paper to reflect that results by Xu et al. and Zhang et al. were achieved on a different model architecture.
>
> **Q: Table 1 should compare CROWN-IBP to Mao et al. [3] with the use of loss fusion.**
>
> Our goal was to compare our results to the best results from the literature regarding certifiability which were yielded by Mao et al. [3] However, our analysis shows that on benchmarks with fewer classes such as MNIST and CIFAR-10, the community should employ CROWN-IBP without loss fusion since this setting yields strictly more favorable trade-offs. In a revised version of the manuscript, we will indicate more clearly that Mao et al. employed loss fusion and our study did not.
>
> **Q: Table 1 should only compare to Mao et al. [3], not to other publications.**
>
> While results by Mao et al. [3] exhibit strongest certified accuracies, the original publications often presented results tuned to another trade-off that also placed importance on performance on clean data. To demonstrate that our method yields strictly better trade-offs than all previously known configurations, we compare against two different values from the literature.

---

> ### Author Response · Authors · 2025-11-20
> **Rebuttal [2]**
>
> **Q: Why does Table 1 compare two different numbers for the comparison against Mao et al. [3]?**
>
> As the reviewer suspected, Table 1 compares two different trade-offs from the literature against two points on the Pareto front produced by our optimisation method. Importantly, in principle, any point on the Pareto front can be selected depending on the requirements at hand. We specifically chose the points to compare favorably with known results in order to demonstrate that our method achieves better trade-offs than previously reported.
>
> **Q: There should be a comparison with a fixed random seed for the training of the models.**
>
> We regret that the paper may not have been sufficiently clear in this regard. For each of the three optimisation runs in our method, we use a fixed seed for training the models. However, due to the exploratory nature of Bayesian optimisation, which may potentially over-explore local optima, it is an important design choice to perform three separate runs to ensure state-of-the-art results.
>
> **Q: Related work is missing.**
>
> We thank the reviewer for their detailed pass and appreciate the comments on the covered related work. We will incorporate the suggested literature in a revised version.
>
> **Q: Bayesian optimisation is not easily parallelisable.**
>
> Parallelisable Bayesian optimisation is an active area of research (see, e.g., [4,5,6]), and with some engineering effort, our method could be adapted to run in a distributed fashion across multiple GPUs. Moreover, the three optimisation runs in our method could each be executed on separate GPUs.
>
> **Q: The paper should discuss the reasons for the stronger trade-offs in more detail.**
>
> We appreciate the reviewer’s interest in a thorough exploration of the design choices identified by our method. We refer the reviewer to Appendix A for a hyperparameter importance analysis conducted on CIFAR-10, which highlights the crucial design choices that led to new state-of-the-art results. In a revised version of the manuscript, we will incorporate these results into the main paper and extend them with an analysis of the hyperparameter configurations discovered on TinyImageNet. Regarding the results on TinyImageNet using MTL-IBP, which achieved new state-of-the-art certified accuracy, the configurations identified by our method exhibited a smaller l1​ regularisation weight and a larger $\epsilon$ factor during training as well as during the PGD attack, compared to the configurations reported by CTBench [3].
>
> **Q: Grid search does not require expertise.**
>
> We respectfully disagree with the reviewer. In grid search, the parameter choices evaluated during hyperparameter optimisation need to be carefully selected to avoid an exhaustive search, since the total number of evaluations is the Cartesian product of all parameter choices investigated. Thus, expertise in choosing appropriate values for the grid search is required. This is evidenced by CTBench [3], which uses different search spaces for each investigated benchmark, carefully constructed based on the authors’ expertise and previous publications. In contrast, we employ the same search space across all benchmarks, demonstrating that our method can be easily adapted to new benchmarks.
>
> **Q: TAPS [7] should be included into the evaluation.**
>
> We refer the reviewer to Appendix E.3, where we explain why the hyperparameters of TAPS, such as the choice of network split, make it particularly difficult to optimise these methods. In addition, while it is true that TAPS performed best on MNIST in CTBench [3], all training methods investigated by CTBench achieved comparable certified accuracies on this dataset. Our analysis in Appendix B further reveals that the best results on MNIST with ε = 0.3 were obtained by the conceptually simplest vanilla IBP method, with only minor performance differences compared to other methods. Therefore, we conclude that on the MNIST benchmark, all IBP-based methods roughly converge to the same performance, and particularly strong results on this dataset do not constitute a compelling argument for the inclusion of any specific method.

---

> ### Author Response · Authors · 2025-11-20
> **Rebuttal [3]**
>
> **Q: How are discrete variables handled by the Gaussian process?**
>
> We use ordinal encoding to represent discrete variables, as is commonly done in Bayesian optimisation [8].
>
> Again, we want to thank the reviewer for their thoughtful questions and detailed feedback and are happy to discuss any further questions that might arise.
>
> [1] Zhang et al., Towards stable and efficient training of verifiably robust neural networks, ICLR 2020
>
> [2] Xu et al., Automatic perturbation analysis for scalable certified robustness and beyond, NeurIPS 2020
>
> [3] Mao et al., CTBench: A Library and Benchmark for Certified Training, ICML 2025
>
> [4] Daulton et al., Differentiable Expected Hypervolume Improvement for Parallel Multi-Objective Bayesian Optimization, NeurIPS 2020
>
> [5] Snoek et al., Practical Bayesian Optimization of Machine Learning Algorithms, NeurIPS 2012
>
> [6] Ginsbourger et al., A Multi-points Criterion for Deterministic Parallel Global Optimization based on Gaussian Processes
>
> [7] Mao et al., Connecting Certified and Adversarial Training, NeurIPS 2023
>
> [8] Cowen-Rivers et al., HEBO: Pushing The Limits of Sample-Efficient Hyperparameter Optimisation, JAIR 70, 2021

---

> > ### Comment · Reviewer_ECKU · 2025-11-23
> >
> > Thanks for the new information from the authors. I will focus on the more important questions in the following.
> >
> > 1. After reading all initial reviews, one issue became clear: **every reviewer thinks this work lacks novelty in that it simply applies BO to certified training**, without new methodologies. In the rebuttal, the authors argued that the main novelty comes from successfully balancing the two conflicting objectives that arise from certified training: clean accuracy and robust accuracy. However, there are works in BO that tackles multi-objective BO as well, e.g., [1]. In particular, such works also apply hyper-volume maximization, similar to the approach in this work. Therefore, I am still not convinced that this work has sufficient novelty as a methodology paper. Note that even the benchmark paper heavily referenced by this work, CTBench, additionally contributes a new library and many new insights into certified training, as well as a structrued hyperparameter tuning and fair comparison.
> >
> > 2. Regarding parallism, the only mentioned parallism is to parallize over different random seeds. However, this is essentially a grid search over random seeds. I did not see how the main algorithm in the BO can be parallelized in this specific setting. In particular, if parallized, how should the next evaluation setting be chosen? If this then simply becomes a "batched" suggestion from the known BO points, then it is simply a new grid search at each step, and no full parallism is possible even given infinite resources.
> >
> >
> > Reference
> >
> > [1] https://arxiv.org/abs/2109.10964

---

### Official Review · Reviewer_8T6c · 2025-10-20

**Soundness:** 3
**Presentation:** 3
**Contribution:** 2
**Rating:** 2
**Confidence:** 3

**Summary:**

The authors present a multi-objective hyperparameter optimization method for certified training methods. Their method generates a Pareto frontier of hyperparameters that address the accuracy/robustness trade-off for certified training methods. They show empirically that some certified training methods can in fact be improved via better hyperparameter tuning, and that different methods have complementary strengths when it comes to optimizing natural versus adversarial performance metrics.

**Strengths:**

-- The paper is well-written and mathematically sound in its descriptions

-- I find the paper easy to follow and well-structured

-- The paper makes empirical contributions that are useful for those in the field of certified training

-- Appreciate that the code is made openly available

**Weaknesses:**

-- My main issue with the paper is a lack of methodological contribution. Bayesian optimization is an out-of-the-box hyperparameter optimization method. The authors apply it to optimize hyperparameters in certified training, but that itself is not a methodological contribution in my opinion

-- Hyperparameter tuning *is* an issue in certified training, but I would prefer to see lighter-weight hyperparameter selection methods compared to doing a smarter grid search. For example, how might one determine the optimal the choice of $\epsilon$ a-priori, to guarantee a certain robustness level, while having good natural accuracy?

**Questions:**

-- I would request that the authors please clarify the novelty of the approach. I would like to see this novelty spelled out more clearly.

-- Does the Bayesian optimization use any structure from the certified training problem? Or is it the generic method of Bayesian optimization (it seems from my reading it is).

---

> ### Author Response · Authors · 2025-11-20
> **Rebuttal [1]**
>
> We thank the reviewer for their insightful and constructive review. We are encouraged by the fact that the reviewer found our paper to be “well-written”, “mathematically sound” and “well-structured” and that the reviewer acknowledged our contribution as “useful for those in the field of certified training”.
>
> Below, we discuss the potential weaknesses highlighted by the reviewer.
>
> **Q: Novelty of the contribution is not spelled out clearly.**
>
> We regret that the paper may not have been sufficiently clear in this regard, and we will improve the presentation of our contributions in a revised version. Automated hyperparameter optimization methods cannot be directly applied to IBP-based certified training due to conflicting objectives between certifiability and standard performance, the computational intractability of complete verification for each configuration, and the absence of expert-designed search spaces. We address these challenges and make the first successful application of automated hyperparameter optimisation in this context. More specifically, we employ multi-objective optimisation, which allows the optimizer to distinguish qualitative differences between configurations that achieve the same sum of objectives. We show that certifiability measured using incomplete verification serves as an efficient, low-cost proxy for identifying best-performing configurations. Additionally, we provide a novel search space that jointly incorporates both general deep learning hyperparameters and method-specific parameters. Furthermore, we introduce a novel evaluation method for certified training by comparing Pareto fronts, providing a nuanced assessment of the trade-offs achievable across the entire objective space. In conclusion, we believe that our paper makes several important contributions, as evidenced by their empirical effectiveness in achieving substantially stronger trade-offs across nearly all investigated scenarios.
>
> **Q: The proposed method is too resource intensive.**
>
> First and foremost, we used 300 evaluations in our experiments but this number can be adjusted based on the compute budget available. We will extend the Appendix with additional experiments that investigate how our method performs when using less trials. Nevertheless, IBP-based certified training has a substantial number of adjustable hyperparameters that require tuning. In a recent benchmarking study by Mao et al. [1], the authors employed grid search over a subset of all parameters we considered using values selected based on previous work and expert knowledge that, thus, do not generalise well to new experimental setups. In this, the authors required 252 evaluations for MTL-IBP on CIFAR-10, $\epsilon=\frac{2}{255}$ which is a similar number in comparison to the 300 evaluations employed in our study. Unfortunately, other studies that relied on manual tuning did not disclose the number of conducted evaluations. In addition, to make the tuning process more efficient and to fully exploit the capabilities of modern GPUs, we used a batch size of 512 in comparison to batch sizes of 128 and 256 used in previous studies. While it was previously thought that larger batch sizes hurt performance, we are the first to show that larger batch sizes can be used when hyperparameters are carefully tuned. Lastly, as outlined in the paper, exploring how more efficient hyperparameter optimisation techniques such as multi-fidelity optimisation could be incorporated would indeed be valuable. However, this is non-trivial, because the warm-up and ramp-up phases used in certified training make intermediate comparisons before training has completed unreliable. Nevertheless, the contributions presented in our submission offer a solid foundation on which future research in this direction can build.
>
> **Q: How could one determine the optimal choice of $\epsilon$ to guarantee a certain trade-off?**
>
> We agree that this direction of research would be both interesting and valuable. However, we consider it out of scope for the present work. Our goal is to demonstrate that smart automated hyperparameter optimisation yields substantially better trade-offs than previous work across standard benchmarks used in the certified training literature, all of which employ a fixed $\epsilon$.
>
> **Q: Is standard Bayesian optimisation used or are there changes to the method?**
>
> At the core of our method, we employ standard Bayesian optimisation. We do not impose restrictions on the choice of the optimiser since the goal of the paper is to enable automated hyperparameter optimisation for certified training in the first place. We consider this a strength of our method, as the underlying optimiser can be easily replaced when new state-of-the-art techniques in automated hyperparameter optimisation become available.
>
>
> [1] Mao et al., CTBench: A Library and Benchmark for Certified Training, ICML 2025

---

> > ### Comment · Reviewer_8T6c · 2025-11-26
> >
> > Dear Authors,
> >
> > I appreciate the detailed response, but as noted by other reviewers, I cannot see the novelty in the procedure, which seems to me to be a Bayesian search method applied to adversarial training hyperparameter selection. This is a standard algorithm for anyone with basic ML/stats training, and while it may be valuable and useful, it is not a methodological contribution on its own.
> >
> > For that reason, I am keeping my score for now.
> >
> > Thank you again for your response.

---

### Official Review · Reviewer_ckqq · 2025-10-27

**Soundness:** 3
**Presentation:** 3
**Contribution:** 1
**Rating:** 2
**Confidence:** 4

**Summary:**

This paper addresses the core challenge of deterministic certified training—balancing verifiable robustness with high clean accuracy. The authors propose an automated multi-objective hyperparameter optimization framework that simultaneously optimizes clean and certified accuracy to generate a full Pareto front. The framework leverages multi-objective Bayesian optimization (EHVI) for efficient hyperparameter search, uses IBP/CROWN-IBP as low-cost proxies, applies constraint optimization to avoid degenerate solutions, and refines results via single-link clustering and αβ-CROWN verification. Experiments on CIFAR-10 and Tiny ImageNet show that the method achieves a better robustness–accuracy trade-off than existing approaches (e.g., MTL-IBP, SABR, CROWN-IBP), offering a more comprehensive view of certified training performance.

**Strengths:**

1.The paper is well-format and easy to read.
2.The code has been anonymously released as open source.

**Weaknesses:**

1.This paper applies hyperparameter optimization to deterministic certified training. However, it contributes little technically to either hyperparameter optimization or deterministic certified training itself.
2.Regarding deterministic certified training, it is important to note that discussions about AI should first focus on usability (i.e., model performance), followed by robustness, fairness, and safety. If an AI model is not practically usable, then the rest of the discussion becomes meaningless.
3.The major issue with deterministic certified training is the significant drop in accuracy, which makes the trained models impractical for real-world use. The experiments in this paper also confirm this limitation: although the proposed approach achieves some improvement over baselines, the accuracy still drops to around 50+% on CIFAR-10 (perturbation radius 2/255) and 20+% on ImageNet (perturbation radius 1/255). Moreover, prior theoretical work has already established an upper bound for deterministic certified training, e.g., for CIFAR-10, this bound is 67.49%.
4.In terms of structure, Chapters 2 and 3 occupy disproportionately large portions of the paper and could be simpler, or even merged into a single section for conciseness.
5.In the experiments, in addition to reporting certified accuracy, it is recommended to include empirical adversarial attack success rates. Furthermore, for model training, runtime efficiency is also critical metrics, and corresponding experiments should be added.

**Questions:**

My main concern is that deterministic certified training has been shown to significantly degrade accuracy, making the resulting models impractical for real-world use. Therefore, I do not think I will change my opinion.

---

> ### Author Response · Authors · 2025-11-20
> **Rebuttal [1]**
>
> We thank the reviewer for their thoughtful review and appreciate that they found the paper “well-formatted and easy to read” and acknowledged that our proposed method achieved better trade-offs than previously known.
>
> We proceed to address the potential weaknesses outlined by the reviewer.
>
> **Q: Certified training has a conceptual issue, i.e. strong decreases in natural performance, which renders work on it unnecessary.**
>
> We find it unfortunate that the reviewer’s main criticism concerns a general challenge inherent to the entire research field of training for certified robustness, rather than an issue specific to our contribution. We agree with the reviewer that IBP-based training methods are currently not suitable for real-world vision applications, as they suffer from a substantial drop in clean accuracy caused by the robustness–accuracy trade-off (see, e.g., [1,2]). Furthermore, we are aware that there exist theoretical upper bounds on achievable certifiable guarantees on commonly employed vision datasets [3]. However, we respectfully disagree that these considerations imply the research area should be abandoned, as the reviewer appears to suggest. First and foremost, the main motivation of ongoing research in the field of IBP-based certified training is to mitigate the impact of the robustness-accuracy trade-off by producing models that, both, perform well on clean data and offer certifiable guarantees to,  ultimately, produce models that are suitable for real-world use. This is testified by, both, IBP-based certified training as well as deterministic neural network verification being active areas of research regularly published at highly relevant conferences such as ICLR, ICML and NeurIPS (see, e.g. [4,5,6,7,8,9]). The present submission offers valuable contributions to that area in (1) enabling the use of automated hyperparameter optimisation to identify best trade-offs on a Pareto front (2) uncovering that better trade-offs, especially with regard to performance on clean data, can be achieved when hyperparameters are carefully tuned and (3) establishing a new evaluation for certified training that takes into account performance on clean data as well as certifiability. With these new perspectives, this work represents, in our opinion, a valuable contribution to training well-performing, certifiably robust, neural networks. Lastly, even if one adopts a pessimistic view regarding the future of deterministic certified training due to theoretical upper bounds, it is important to recognize that many real-world applications requiring deterministic guarantees, such as medical diagnostics, aerospace, or energy systems, typically involve neural networks with far fewer inputs than those in the vision domain. In these settings, IBP-based techniques can produce high-performing, certifiably robust networks with tangible real-world impact. Unfortunately, the community has largely focused on vision benchmarks for evaluating certified training methods, while realistic application benchmarks remain publicly unavailable.
>
> **Q: The work contributes little to hyperparameter optimisation or certified training.**
>
> We regret that the paper might not have been clear in highlighting our proposed contributions. We agree that, ultimately, we do not propose conceptual changes to certified training and automated hyperparameter optimisation. However, we propose multiple adaptations that enable hyperparameter optimisation for IBP-based certified training in the first place. These include a multi-objective optimisation approach that enables the optimiser to recognize qualitative differences between configurations that achieve similar sums of certified and natural accuracy, the use of low-cost proxies for the certifiability objective during optimisation and general hyperparameter search spaces that encompass both, general deep learning hyperparameters as well as method-specific hyperparameters. With these contributions, we are the first to successfully apply automated hyperparameter optimisation to IBP-based certified training, achieving substantial performance improvements. Moreover, we introduce a new perspective on evaluating certified training methods by analysing their performance through Pareto fronts.

---

> ### Author Response · Authors · 2025-11-20
> **Rebuttal [2]**
>
> **Q: Empirical adversarial attack success rates are not reported.**
>
> We agree that adversarial attack success rates offer an additional important dimension to the evaluation of certified training approaches and will add empirical success rates to Table 1 in a revised version of the manuscript. The updated table with adversarial success rates is included below:
>
> | Dataset        | ε            | Method     | Clean Acc. [%] (Lit.) | Cert. Acc. [%] (Lit.) | Clean Acc. [%] (ours) | Cert. Acc. [%] (ours) | Adv. Acc. [%] (ours) |
> |----------------|--------------|------------|------------------------|------------------------|------------------------|-------------------------|------------------------|
> | **CIFAR-10**   | 2/255        | MTL-IBP    | 80.11                  | 63.24                  | 79.97                  | **63.99**               | 69.12                 |
> |                |              |            | 78.82                  | 64.41                  | **79.87**              | 64.54                  | 69.44                 |
> |                |              | SABR       | 79.24                  | 62.84                  | **81.95**              | **64.11**               | 69.87                 |
> |                |              |            | 77.86                  | 63.61                  | **80.15**              | **64.44**               | 68.01                 |
> |                |              | IBP        | 66.84                  | 52.85                  | **71.39**              | **55.54**               | 55.97                 |
> |                |              |            | 67.49                  | 55.99                  | **69.37**              | 55.62                  | 55.94                 |
> |                |              | CROWN-IBP  | 71.52                  | 53.97                  | **77.44**              | **59.25**               | 60.99                 |
> |                |              |            | 67.60                  | 57.11                  | **75.70**              | **60.39**               | 62.54                 |
> |                | 8/255        | MTL-IBP    | 53.35                  | **35.44**               | **55.25**              | 34.49                  | 35.56                 |
> |                |              |            | 54.28                  | 35.41                  | 54.18                  | 35.27                  | 36.21                 |
> |                |              | SABR       | 52.38                  | 35.13                  | **54.93**              | 34.96                  | 35.91                 |
> |                |              |            | 52.71                  | **35.34**               | **56.06**              | 34.26                  | 35.52                 |
> |                |              | IBP        | 48.94                  | 34.97                  | **52.62**              | 35.09                  | 35.56                 |
> |                |              |            | 48.51                  | 35.28                  | **51.02**              | 35.35                  | 35.92                 |
> |                |              | CROWN-IBP  | 46.29                  | 33.38                  | **55.11**              | 33.77                  | 34.58                 |
> |                |              |            | 48.25                  | 32.59                  | **52.47**              | **34.41**               | 34.87                 |
> | **Tiny ImageNet** | 1/255     | MTL-IBP    | 37.56                  | 26.09                  | **39.80**              | **30.45**               | 32.62                 |
> |                |              |            | 35.97                  | 27.73                  | **39.75**              | **30.67**               | 32.74                 |
> |                |              | SABR       | 28.85                  | 20.46                  | **40.61**              | **28.86**               | 31.69                 |
> |                |              |            | 30.58                  | 20.96                  | **42.10**              | **26.38**               | 31.62                 |
> |                |              | IBP        | 25.92                  | 17.87                  | **34.24**              | **20.03**               | 21.68                 |
> |                |              |            | 26.77                  | 19.82                  | **32.12**              | **21.53**               | 22.35                 |
> |                |              | CROWN-IBP  | 25.62                  | 17.93                  | **32.38**              | **20.72**               | 21.40                 |
> |                |              |            | 28.44                  | 22.14                  | **30.82**              | 22.20                  | 22.68                 |

---

> ### Author Response · Authors · 2025-11-20
> **Rebuttal [3]**
>
> **Q: Chapters 2 and 3 should be made more concise.**
>
> We appreciate the reviewer's thoughts on the structure of the paper and will shorten Background and Related Work sections in a revised version of the manuscript.
>
> **Q: Running time is not reported.**
>
> We refer the reviewer to Appendix D of the paper where we discuss computational costs of the optimisation procedure as well as the computational costs of the verification.
>
> Again, we want to thank the reviewer for their thoughtful questions and detailed feedback and are happy to discuss any further questions that might arise. Ultimately, we hope that the reviewer can assess the contribution and importance of the present submission as an addition to an important subfield of deep learning, even when they personally do not believe in its success.
>
> [1] Tsipras et al., Robustness May Be at Odds with Accuracy, ICLR 2019
>
> [2] Zhang et al., Theoretically principled trade-off between robustness and accuracy, ICML 2019
>
> [3] Zhang et al., Certified Robust Accuracy of Neural Networks Are Bounded due to Bayes Errors, CAV 2024
>
> [4] De Palma et al., Expressive Losses for Verified Robustness via Convex Combinations, ICLR 2024
>
> [5] Mao et al., Understanding Certified Training via Interval Bound Propagation, ICLR 2024
>
> [6] Mao et al., CTBench: A Library and Benchmark for Certified Training, ICML 2025
>
> [7] Zhou et al., Scalable Neural Network Verification with Branch-and-Bound Inferred Cutting Planes, NeurIPS 2024
>
> [8] Chiu et al., SDP-CROWN: Efficient Bound Propagation for Neural Network Verification with Tightness of Semidefinite Programming, ICML 2025
>
> [9] Zhou et al., Clip-and-Verify: Linear Constraint-Driven Domain Clipping for Accelerating Neural Network Verification, NeurIPS 2025

---

> ### Comment · Reviewer_ckqq · 2025-11-25
>
> After reading all the reviews and response from authors, I insist the original decision, since the paper has limited contributions. For the practicality of deterministic certified training, the authors claim that it may work better on lower-dimensional datasets, likely based on the intuitive assumption that the drop in certified accuracy is inversely correlated with input dimensionality, a trend that is indeed common in many deep learning metrics. Unfortunately, this intuition does not hold in practice. For some simple tabular datasets, the degradation in certified accuracy is even more severe than that observed on image datasets.

---

### Official Review · Reviewer_h7XM · 2025-10-27

**Soundness:** 3
**Presentation:** 4
**Contribution:** 2
**Rating:** 2
**Confidence:** 4

**Summary:**

The authors merge the concepts of robustness (via interval bound propagation) and hyperparameter optimization, in an attempt to improve performance of IBP style methods.

**Strengths:**

On the whole, this was a well written (there'll be a few notes on this below) paper that works in a space that is relevant. The authors align with community expectations with regards to testing datasets (typically now I'd prefer to see Imagenet rather than Tiny Imagenet, however I appreciate that this isn't viable in the IBP side of the certified robustness community).

Fundamentally too, the idea of articulating pareto fronts (relative to single points in hyperparameter space) and fundamentally there is a need for, as the authors say, "fair, principled and nuanced comparison of the performance of different methods. We show that most methods yield better trade-offs".

**Weaknesses:**

Unfortunately, while it's well written, it's also relatively difficult to see impact here. The authors have merged two known techniques - hyperparameter optimization and IBP - and have measured some results. This was clearly computationally challenging, in terms of the resources required, but as far as I can see this is a paper that mainly says "parameters could be slightly improved". But doing this requires huge investments of computational resources, and is still, fundamentally, while also being inherently tied to the very same small architectures that inherently hamstring IBP style techniques.

Fundamentally, the idea of hyperparameter optimisation of IBP style techniques feels like rearranging deckchairs on the Titanic - yielding small improvements for something that feels a tiny bit doomed. This is a bit of a harsh perspective on IBP style techniques, but their GPU and architectural limitations are significant, and performing Prateo optimisation on something that requires significant resource investments only makes the problem worse. I do not see how there would lead to any real world take away from this. Either 1) someone is following the exact same experimental setup, in which case the updated hyperparameters only form as an updated point of reference, but with limited other utility or 2) someone is using a different experimental set up, in which case they could independently use hyperparameter optimization without any need to reference the authors work.

Results were also not contexutalised with uncertainties or error bounds, which are almost certainly non-trivial given the sample size of 300.

The following comments are more minor, and primarily relate to writing quality.
L12 - robustness not defined, assumed knowledge
L15 - "substantial computation requirements" - why is this true? Yes, to someone in the field this is clear, but the abstract is making strong assumptions about who the reader is.
L15 - but don't certified training ethods also require significant computational investment?
L38 - "map generation for autonomous driving" - how is this a safety critical task? Where are the risks?
L39: Adversarial example sentence is a non sequitur - as presented it disguises the impact of these on safety critical problems.
L42 How do these "play an important role in diagnosing weaknesses" - finding an adversarial example is no guarantee that no smaller / better AE could be found by an adversary?
L46-48: Are these network wide safty properties, or sample wise properties?
L50: Why is L_inf popular?
L62: SOTA is not IBP. Randomised smoothing has, arguably, surpassed it, as IBP imposes significant GPU ram requirements that limit its use to small networks (and also prevents the use of some architectures), whereas RS only incurs a time penalty, with almost 0 memory penalty, meaning that it can be applied to a broader range of network architectures.
L71: "Moreover, these methods require tuning additional hyperparameters, such as the learning rate and the number of warm-up epochs, which strongly influence training stability and final performance" - how is this any different to any neural network? What is different here besides the application of fine tuning to a new domain?
L144: The length of this sentence causes problems. Madry's specific contribution was around the worst case loss in the l_inf norm ball, but the way this is phrased that part is 3 lines deep, and quite disconnected from the context that originally establishes that MMadry is the first to introduce...
Table 1 isn't formatted in a way that allows for easy comparisons, in that it goes clean acc, cert acc, clean acc, cert acc.
Figure 1 is apparently a comparison between your work and Mao's, but it's not clear which is yours and which is Mao's in the majority of f  figures. , nor how things should be compared.
Figure 2 "when prioritising natural accuracy" is a very....loose statement. One could prioritise natural accuracy but still care about certified accuracy, in which case they may choose MTL, or possibly even SABR, depending upon their threshold for natural accuracy?
Figures 3 and 4 are effectively unreadable. This style of presentation is odd, to say the least. And some words are cut off.
Certified is a descriptor of a technique generally, and should be treated as a proper noun - including in the bibliography.
Adam is ADAM - bibliography.
Batch Normalization is a proper noun - bibliography.
Deep Neural Networks and Random Forests - see above.

**Questions:**

How would this scale to non l_inf boundings?

What is the value of hyperparameter optimisation in a system which is incredibly expensive from a computational perspective? What scenarios would allow this to occur?

How would performance scale with increased epsilon?

Are the values in table 1 the values reported by these papers, or were they the results you constructed using the approaches from prior works?

The experimental setup section doesn't make sense to me - why does C10 have different resources avaiable to it than TI/MNIST, and what does line 930 mean relative to lines 923-929. If I'm reading L930 correctly, you used one GPU (either a NVL or SXM H100 of unspecified GPU ram), 120gb of ram, and 24 CPU cores - if so, then what exactly was the point of lines 923-929, besides mentioning what resources you have available? Also why were these details only in the appendices?

---

> ### Author Response · Authors · 2025-11-20
> **Rebuttal [1]**
>
> We want to thank the reviewer for their thoughtful and detailed review. We will incorporate the minor comments on writing quality as well as a clarification of the identified weaknesses into a revised version of the manuscript and appreciate the reviewer for their detailed pass. We are delighted to hear that the reviewer acknowledges our submission as a “well-written paper” that tackles a “relevant” problem. Furthermore, we are encouraged that our proposed evaluation of certified training approaches using Pareto fronts that enable a nuanced and fair comparison was appreciated.
>
> In the following, we address the potential weaknesses of our paper identified by the reviewer.
>
> **Q: IBP-based training is not SOTA.**
>
> We respectfully disagree with the reviewer on this view. While it is true that approaches based on randomised smoothing are compelling in that they scale to larger architectures and datasets and incur a smaller performance penalty than IBP-based techniques, they do ultimately not yield strong, deterministic guarantees but only probabilistic ones. While these guarantees may be sufficient for certain application areas, high-risk domains (e.g. aircraft navigation, medicine, autonomous driving, …) require deterministic guarantees where IBP-style techniques are state of the art. This is testified by, both, IBP-based certified training as well as deterministic neural network verification being active areas of research regularly published at highly relevant conferences such as ICLR, ICML and NeurIPS (see, e.g. [1,2,3,4,5,6]). In addition, smoothed networks yielded by randomised smoothing incur a substantial computational overhead at inference time that makes them unsuitable for real-time applications. Another line of research that offers strong deterministic guarantees are Lipschitz-networks such as SortNet [7]. These approaches yield state-of-the-art results on higher perturbation radii but, fundamentally, cannot surpass IBP-based approaches on smaller radii on CIFAR-10 and TinyImageNet (see, e.g., [1,3,7]). Lastly, Lipschitz-networks require non-standard neural network layers and operations that are generally not supported in popular deep learning libraries and standards, therefore hindering their real-world applicability. Thus, in conclusion, we do not consider IBP-based certified training techniques to be “a tiny bit doomed” and hope that the reviewer can assess our contribution as an addition to an important subfield of certified robustness, even when he personally sees more potential in alternative training paradigms.

---

> ### Author Response · Authors · 2025-11-20
> **Rebuttal [2]**
>
> **Q: Impact is limited and there is no obvious real-world take away.**
>
> We appreciate that the reviewer highlighted that he did not fully grasp the real-world implications of our method and we regret that the manuscript might not have been clear in that regard. We will improve on that in a revised version of the paper. As correctly identified by the reviewer, there are two main scenarios to consider. First, “someone is following the exact same experimental setup” where the reviewer sees our results as an “updated point of reference, but with limited other utility”. We respectfully disagree. Our analysis provides, for the first time, a Pareto front of best-performing configurations, showing that substantially better trade-offs are achievable in most scenarios. This stands in contrast to the single-point comparisons used in previous work, which can be misleading (see Figure 1). In addition, our analysis uncovered performance complementarities (e.g. SABR for stronger natural accuracies and MTL-IBP for stronger certifiable guarantees on CIFAR-10, $\epsilon=\frac{2}{255}$) that can considerably assist practitioners and researchers in choosing the best method for their requirements. In the second scenario described by the reviewer as “someone is using a different experimental setup, in which case they could independently use hyperparameter optimisation without any need to reference the author’s work” also benefits substantially from the proposed contributions. As outlined in Appendix E, we argue that out-of-the-box approaches are not suitable in the context of IBP-based certified training. First, there is no clear opinion in the literature whether one configuration achieving a natural accuracy of 75% and a certified accuracy of 50% should be preferred over a configuration achieving 80% and 45% respectively. In addition, the two considered configurations display substantially different qualities that, we hypothesise, are required to successfully steer the optimiser into well-performing regions. Therefore, we propose to employ state-of-the-art multi-objective Bayesian hyperparameter optimisation and show that these methods can find Pareto-fronts with trade-offs that outperform previous work. In safety-critical real-world applications where every certification matters, our method provides better configurations with considerably less human labour involved than required by manual tuning. Additionally, we demonstrate the effectiveness of employing incomplete methods for the certifiability objective during the optimisation procedure. This modification makes the optimisation computationally tractable in the first place and, thus, is an important finding for different experimental setups. Lastly, we provide expert-designed, general search spaces for all investigated methods that can be used to perform automated hyperparameter optimisation for different experimental setups without any domain-specific knowledge, making our contribution highly-relevant to practitioners.
>
> **Q: The proposed method is too computationally expensive.**
>
> First and foremost, we used 300 evaluations in our experiments but this number can be adjusted based on the compute budget available. We will extend the Appendix with additional experiments that investigate how our method performs when using less trials. Nevertheless, IBP-based certified training has a substantial number of adjustable hyperparameters that require tuning. In a recent benchmarking study by Mao et al. [3], the authors employed grid search over a subset of all parameters we considered using values selected based on previous work and expert knowledge that, thus, do not generalise well to new experimental setups. In this, the authors required 252 evaluations for MTL-IBP on CIFAR-10, $\eps=\frac{2}{255}$ which is a similar number in comparison to the 300 evaluations employed in our study. Unfortunately, other studies that relied on manual tuning did not disclose the number of conducted evaluations. In addition, to make the tuning process more efficient and to fully exploit the capabilities of modern GPUs, we used a batch size of 512 in comparison to batch sizes of 128 and 256 used in previous studies (see, e.g., [1,3]). While it was previously thought that larger batch sizes hurt performance, we are the first to show that larger batch sizes can be used when hyperparameters are carefully tuned.
>
> **Q: Results were not contextualised with uncertainties or error bounds.**
>
> In Table 1, we show selected configurations from the Pareto fronts obtained by our method and in Figures 1 and 2 we show all Pareto-dominant configurations per training method. Thus, each point in the Figures and each line in Table 1 refers to one single configuration out of the 300 evaluated configurations. Therefore, error bounds or uncertainties are not applicable here. In Appendix C.4 we report variances of all configurations in the Pareto fronts over three pseudo-random seeds.

---

> ### Author Response · Authors · 2025-11-20
> **Rebuttal [3]**
>
> **Q: How would this scale to non-$\ell_\infty$ bounds?**
>
> IBP-based certified training is generally only evaluated on $\ell_\infty$ perturbations since propagating a hyperbox enclosure of other norm balls is equivalent to propagating the corresponding $\ell_\infty$ norm ball. However, there is a recent line of research on applying IBP-based certified training to semantic, non $\ell_\infty$-bounded, perturbation such as Bias Fields [8] or perturbations in the frequency domain [9]. Since those methods have similar numbers of parameters, we fully expect our method to perform well in these scenarios too. However, while we would be very interested in exploring these directions in future work, we consider a thorough evaluation in that regard out of scope for the present work.
>
> **Q: Are values in Table 1 taken from the respective publications or reproduced?**
>
> The values are taken from the respective publications as indicated by the “Source” column. In Appendix C.3 we reproduce configurations reported in the literature using the CTRAIN library employed for our experiments to ensure that reported performance gains cannot be attributed to different implementations.
>
> **Q: Why were two different hardware setups used for the experiments?**
>
> Due to the substantial resource requirements of conducting a large-scale study across several training methods, datasets and $\epsilon$ values as presented in the submitted paper, we split up our experiments across two different compute clusters. However, we did not encounter or notice any effects resulting from the small hardware differences during training of the MNIST and TinyImageNet models. We will incorporate the information on the employed hardware into the main paper using the additional page available after acceptance.
>
> Again, we want to thank the reviewer for their thoughtful questions and detailed feedback and are happy to discuss any further questions that might arise.
>
>
> [1] De Palma et al., Expressive Losses for Verified Robustness via Convex Combinations, ICLR 2024
>
> [2] Mao et al., Understanding Certified Training via Interval Bound Propagation, ICLR 2024
>
> [3] Mao et al., CTBench: A Library and Benchmark for Certified Training, ICML 2025
>
> [4] Zhou et al., Scalable Neural Network Verification with Branch-and-Bound Inferred Cutting Planes, NeurIPS 2024
>
> [5] Chiu et al., SDP-CROWN: Efficient Bound Propagation for Neural Network Verification with Tightness of Semidefinite Programming, ICML 2025
>
> [6] Zhou et al., Clip-and-Verify: Linear Constraint-Driven Domain Clipping for Accelerating Neural Network Verification, NeurIPS 2025
>
> [7] Zhang et al., Rethinking Lipschitz Neural Networks and Certified Robustness: A Boolean Function Perspective, NeurIPS 2022
>
> [8] Henriksen et al., Robust Training of Neural Networks against Bias Field Perturbations, AAAI-23
>
> [9] Hanspal et al., Robustness to Perturbations in the Frequency Domain: Neural Network Verification and Certified Training, WACV Workshops 2025

---

> ### Comment · Reviewer_h7XM · 2025-11-20
>
> Quick notes, please don't immediately message back.
>
> "This is testified by, both, IBP-based certified training as well as deterministic neural network verification being active areas of research regularly published at highly relevant conferences such as ICLR, ICML and NeurIPS (see, e.g. [1,2,3,4,5,6]). In addition, smoothed networks yielded by randomised smoothing incur a substantial computational overhead at inference time that makes them unsuitable for real-time applications"
>
> Argument from authority doesn't convince. That research exists in these spaces doesn't obviate my original concern, which is the ability to scale to problems of interest.
>
> As to RS....so you mean IBP doesn't have a substantial computational overhead? I'd be careful with that as an argument. GPU memory limitations, time, etc., are all not exactly cheap. And if something cannot scale to, say, Imagenet sized data, then one could argue that it's computational overhead is infinite. Does RS introduce latency? Certainly. But not sure I'm buying the premise of your argument.
>
> Especially, as you note, the limitations with IBP w.r.t the associated norms. What happens if ones risk profile is not l-inf bounds? Again, the computational overhead becomes infinite, or, from another perspective, moot, because it's a problem that cannot be solved.
>
> "require deterministic guarantees"
>
> Can you point to a paper with a clear risk model, that provides a level of distinction between the level of risk associated with a high probability RS certification, and a bounded IBP style certification? And again, what happens if IBP cannot scale to the problem of interest?
>
> " a Pareto front of best-performing configurations, showing that substantially better trade-offs are achievable in most scenarios." and "Therefore, we propose to employ state-of-the-art multi-objective Bayesian hyperparameter optimisation and show that these methods can find Pareto-fronts with trade-offs that outperform previous work."
>
> Okay, but how would one use this? How generalisable is the Pareto front to other scenarios? How far can you push things? What value does this communicate to someone working in a space that is not closely tied to those that you performed experiments under? How much work would be required to find this again, and when would one need to perform this optimization again? How much utility is there to be gained in the work required for relatively minor improvements? These are the points I was raising in the review.
>
> "This stands in contrast to the single-point comparisons used in previous work, which can be misleading (see Figure 1)"
>
> Don't disagree, but at this point I'm not convinced that you've made the argument that you've fixed the problem. Sure, one could argue that you've taken a step forward, but I would think that there's an opportunity for this to be a better paper if these could be articulated. And more to the point, publishing this may crowd out that better paper from existing. Which is not to make perfect the enemy of the good, but rather to say that I think that conceptually you're moving in the right direction, but I'm not necessarily convinced that where the authors have landed fully satisfies the brief. Willing to be proven wrong of course, but still.
>
> "Due to the substantial resource requirements of conducting a large-scale study across several training methods, datasets and $\epsilon$ values as presented in the submitted paper, we split up our experiments across two different compute clusters."
>
> And this, dear author, is kind of why I asked a lot of the questions above. If your contribution is the process for optimization, I ask how realistic is it for someone to go through this same process, when not doing it will likely lead to a good enough solution, and what is so novel about the process that it's not just standard optimization wearing a new hat? If your contribution is the result of this process, then how extensible / generalisable / useful is it? That I'm asking these questions is at the core of why my review landed where it did.
>
> "We appreciate that the reviewer highlighted that he did not fully grasp "
>
> Interesting phrasing here 1) Because implicitly placing the blame on the reader at first (you clarify this later in the sentence, but still, not a strong start) and 2) because you assume the reader is male. Not sure I'd make either of those choices.

---

### Meta-Review · Area_Chair_yQtr · 2026-01-07

**Summary:**

1. All reviewers highlight that the lack of technical contributions or novelty is the major concern.
2. Some reviewers are pessimistic about this line of research - interval bound propagation, due to its weak certified robustness guarantees under a given compute budget and the significant tradeoff against utility when compared to Lipschitz networks and randomized smoothing. As a result, the incremental performance optimization may not be a significant contribution.
3. Some reviewers are concerned about the compute cost. The evaluation requires around 300 trials per benchmark. The required compute may be too heavy to generalize beyond the evaluated methods.
4. Research findings may be too specific. It may not attract a broader audience. The discovered performance orderings between different relaxation-based methods may not bring much insight for similar domains or broader ML robustness in general.
5. Experiment settings might not be sound enough. Some experiment comparison or illustration may be ambiguous. The limited sample size may weaken the finding.

**Reviewer Concerns:**

After the rebuttal, the concerns 1, 3, and 4 persist. Some reviewers are not strong in insisting the concern 2, given the rebuttal. Concern 5 is mostly addressed. Further, all reviewers agree that the writing quality is high and most parts of the paper are clear.

**Reviewer Scores:**

All reviewers have replied to the author's response, indicating no score change, and no further response from the author is provided. As a result, I would anticipate that scores are already final.

---

### Decision · Program_Chairs · 2026-01-26

Reject